# Learning Distilled Collaboration Graph for Multi-Agent Perception

**Yiming Li**
New York University
yimingli@nyu.edu

**Shunli Ren**
Shanghai Jiao Tong University
renshunli@sjtu.edu.cn

**Pengxiang Wu**
Rutgers University
pxiangwu@gmail.com

**Siheng Chen**\*
Shanghai Jiao Tong University
sihengc@sjtu.edu.cn

**Chen Feng**\*
New York University
cfeng@nyu.edu

**Wenjun Zhang**
Shanghai Jiao Tong University
zhangwenjun@sjtu.edu.cn

## Abstract

To promote better performance-bandwidth trade-off for multi-agent perception, we propose a novel *distilled collaboration graph* (DiscoGraph) to model trainable, pose-aware, and adaptive collaboration among agents. Our key novelties lie in two aspects. First, we propose a teacher-student framework to train DiscoGraph via knowledge distillation. The teacher model employs an early collaboration with holistic-view inputs; the student model is based on intermediate collaboration with single-view inputs. Our framework trains DiscoGraph by constraining post-collaboration feature maps in the student model to match the correspondences in the teacher model. Second, we propose a matrix-valued edge weight in DiscoGraph. In such a matrix, each element reflects the inter-agent attention at a specific spatial region, allowing an agent to adaptively highlight the informative regions. During inference, we only need to use the student model named as the distilled collaboration network (DiscoNet). Attributed to the teacher-student framework, multiple agents with the shared DiscoNet could collaboratively approach the performance of a hypothetical teacher model with a holistic view. Our approach is validated on V2X-Sim $1.0$, a large-scale multi-agent perception dataset that we synthesized using CARLA and SUMO co-simulation. Our quantitative and qualitative experiments in multi-agent 3D object detection show that DiscoNet could not only achieve a better performance-bandwidth trade-off than the state-of-the-art collaborative perception methods, but also bring more straightforward design rationale. Our code is available on https://github.com/ai4ce/DiscoNet.

## 1 Introduction

Perception, which involves organizing, identifying and interpreting sensory information, is a crucial ability for intelligent agents to understand the environment. Single-agent perception [4] has been studied extensively in recent years, *e.g.*, 2D/3D object detection [18, 27], tracking [23, 22] and segmentation [25, 16], *etc.* Despite its great progress, single-agent perception suffers from a number of shortcomings stemmed from its individual perspective. For example, in autonomous driving [8], the LiDAR-based perception system can hardly perceive the target in the occluded or long-range areas. Intuitively, with an appropriate collaboration strategy, multi-agent perception could fundamentally upgrade the perception ability over single-agent perception.

To design a collaboration strategy, current approaches mainly include raw-measurement-based early collaboration, output-based late collaboration and feature-based intermediate collaboration.

---

\*Corresponding authors.

35th Conference on Neural Information Processing Systems (NeurIPS 2021).

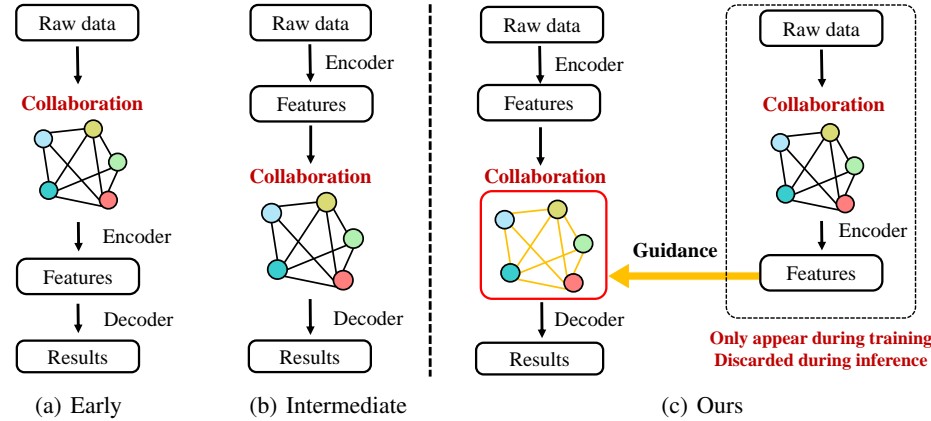

Figure 1: Scheme comparison. (a) Early collaboration requires an expensive bandwidth for raw data transmission. (b) Intermediate collaboration needs an appropriate collaboration strategy. (c) The proposed method incorporates both early and intermediate collaboration into a knowledge distillation framework, enabling the knowledge of early collaboration to guide the training of an intermediate collaboration strategy, leading to better trade-off between performance and bandwidth.

Early collaboration [3] aggregates the raw measurements from all the agents, promoting a holistic perspective; see Fig. 1 (a). It can fundamentally solve the occlusion and long-range issues occurring in the single-agent perception; however, it requires a lot of communication bandwidth. Contrarily, late collaboration aggregates each agent's perception outputs. Although it is bandwidth-efficient, each individual perception output could be noisy and incomplete, causing unsatisfying fusion results. To deal with the performance-bandwidth trade-off, intermediate collaboration [19, 34, 20] has been proposed to aggregate intermediate features across agents; see Fig. 1 (b). Since we can squeeze representative information to compact features, this approach can potentially both achieve communication bandwidth efficiency and upgrade perception ability; however, a bad design of collaboration strategy might cause information loss during feature abstraction and fusion, leading to limited improvement of the perception ability.

To achieve an effective design of intermediate collaboration, we propose a distilled collaboration graph (DiscoGraph) to model the collaboration among agents. In DiscoGraph, each node is an agent with real-time pose information and each edge reflects the pair-wise collaboration between two agents. The proposed DiscoGraph is trainable, pose-aware, and adaptive to real-time measurements, reflecting dynamic collaboration among agents. It is novel from two aspects. First, from the training aspect, we propose a teacher-student framework to train DiscoGraph through knowledge distillation [1, 10, 26]; see Fig. 1 (c). Here the teacher model is based on early collaboration with holistic-view inputs and the student model is based on intermediate collaboration with single-view inputs. The knowledge-distillation-based framework enhances the training of DiscoGraph by constraining the post-collaboration feature maps in the student model to match the correspondences in the teacher model. With the guidance of both output-level supervision from perception and feature-level supervision from knowledge distillation, the distilled collaboration graph promotes better feature abstraction and aggregation, improving the performance-bandwidth trade-off. Second, from the modeling aspect, we propose a matrix-valued edge weight in DiscoGraph to reflect the collaboration strength with a high spatial resolution. In the matrix, each element represents the inter-agent attention at a specific spatial region. This design allows the agents to adaptively highlight the informative regions and strategically select appropriate partners to request supplementary information.

During inference, we only need to use the student model. Since it leverages DiscoGraph as the key component, we call the student model as the distilled collaboration network (DiscoNet). Multiple agents with the shared DiscoNet could collaboratively approach the performance of a hypothetical teacher model with the holistic view.

To validate the proposed method, we build V2X-Sim 1.0, a new large-scale multi-agent 3D object detection dataset in autonomous driving scenarios based on CARLA and SUMO co-simulation platform [6]. Comprehensive experiments conducted in 3D object detection [37, 28, 29, 17] have shown that the proposed DiscoNet achieves better performance-bandwidth trade-off and lower communication latency than the state-of-the-art intermediate collaboration methods.

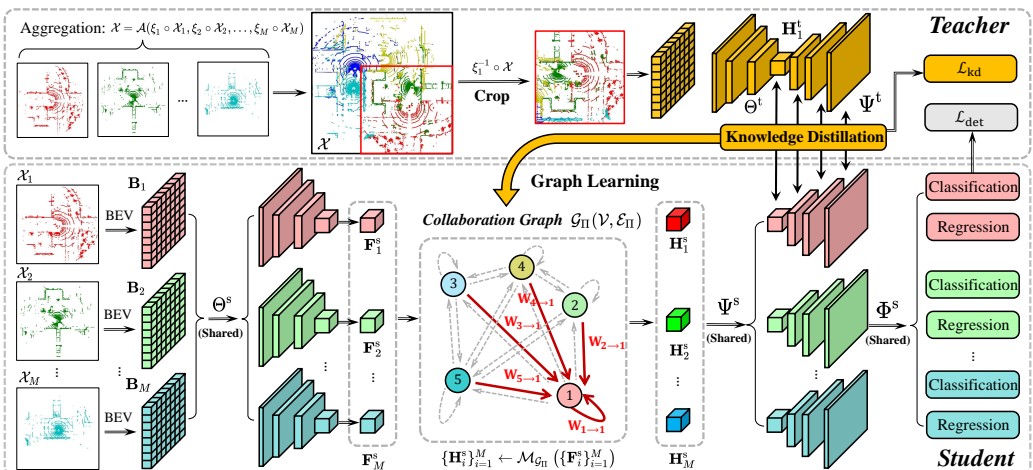

Figure 2: Overall teacher-student training framework. In the student model, for Agent 1 (in red), its input single-view point cloud can be converted to a BEV map, and then be consumed by a shared encoder $\Theta^s$ to obtain the feature map $\mathbf{F}_i^s$. Based on the collaboration graph $\mathcal{G}_\Pi$, Agent 1 aggregates the neural messages from other agents and obtains the updated feature map $\mathbf{H}_i^s$. The shared header following the shared decoder outputs the 3D detection results. In the teacher model, we aggregate the points collected from all agents to obtain a holistic-view point cloud. We crop the region and align the pose to ensure the BEV maps in both teacher and student models are synchronized. We constrain all the post-collaboration feature maps in the student model to match the correspondences in the teacher model through knowledge distillation, resulting in a collaborative student model.

## 2 Multi-Agent Perception System with Distilled Collaboration Graph

This work considers a collaborative perception system to perceive a scene, where multiple agents can perceive and collaborate with each other through a broadcast communication channel. We assume each agent is provided with an accurate pose and the perceived measurements are well synchronized. Then, given a certain communication bandwidth, we aim to maximize the perception ability of each agent through optimizing a collaboration strategy. To design such a strategy, we consider a teacher-student training framework to integrate the strengths from both early and intermediate collaboration. During training, we leverage an early-collaboration model (teacher) to teach an intermediate-collaboration model (student) how to collaborate through knowledge distillation. Here the teacher model and the student model are shared by all the agents, but each agent would input raw measurements from its own view to either model; see Fig. 2. During inference, we only need the student model, where multiple agents with the shared student model could collaboratively approach the performance of the teacher model with the hypothetical holistic view.

In this work, we focus on the perception task of LiDAR-based 3D object detection because the unifying 3D space naturally allows the aggregation of multiple LiDAR scans. Note that in principle the proposed method is generally-applicable in collaborative perception if there exists a unified space to aggregate raw data across multiple agents.

### 2.1 Student: Intermediate collaboration via graphs

**Feature encoder.** The functionality is to extract informative features from raw measurements for each agent. Let $\mathcal{X}_i$ be the 3D point cloud collected by the $i$th agent ($M$ agents in total), the feature of the $i$th agent is obtained as $\mathbf{F}_i^s \leftarrow \Theta^s(\mathcal{X}_i)$, where $\Theta^s(\cdot)$ is the feature encoder shared by all the agents and the superscript s reflects the student mode. To implement the encoder, we convert a 3D point cloud to a bird's-eye-view (BEV) map, which is amenable to classic 2D convolutions. Specifically, we quantize the 3D points into regular voxels and represent the 3D voxel lattice as a 2D pseudo-image, with the height dimension corresponding to image channels. Such a 2D image is virtually a BEV map, whose basic element is a cell that is associated with a binary vector along the vertical axis. Let $\mathbf{B}_i \in \{0,1\}^{K \times K \times C}$ be the BEV map of the $i$th agent associated with $\mathcal{X}_i$. With this map, we can apply four blocks composed of 2D convolutions, batch normalization and ReLU activation to gradually reduce the spatial dimension and increase the number of channels for the BEV, to obtain the feature map $\mathbf{F}_i^s \in \mathbb{R}^{\bar{K} \times \bar{K} \times \bar{C}}$ with $\bar{K} \times \bar{K}$ the spatial resolution and $\bar{C}$ the number of channels.

**Feature compression.** To save the communication bandwidth, each agent could compress its feature map prior to transmission. Here we consider a $1 \times 1$ convolutional autoencoder [24] to compress/decompress the feature maps along the channel dimension. The autoencoder is trained together with the whole system, making the system work with limited collaboration information.

**Collaboration graph process.** The functionality is to update the feature map through data transmission among the agents. The core component here is a *collaboration graph*[1] $\mathcal{G}_{\Pi}(\mathcal{V}, \mathcal{E}_{\Pi})$, where $\mathcal{V}$ is the fixed node set and $\mathcal{E}_{\Pi}$ is the trainable edge set. Each node in $\mathcal{V}$ is an agent with the real-time pose information; for instance, $\boldsymbol{\xi}_i \in \mathfrak{se}(3)$ is the $i$th agent's pose in the global coordinate system; and each edge in $\mathcal{E}_{\Pi}$ is trainable and models the collaboration between two agents, with $\Pi$ an edge-weight encoder, reflecting the trainable collaboration strength between agents. Let $\mathcal{M}_{\mathcal{G}_{\Pi}}(\cdot)$ be the collaboration process defined on the collaboration graph $\mathcal{G}_{\Pi}$. The feature maps of all the agents after collaboration are $\{\mathbf{H}_i^{\mathrm{s}}\}_{i=1}^M \leftarrow \mathcal{M}_{\mathcal{G}_{\Pi}}\left(\{\mathbf{F}_i^{\mathrm{s}}\}_{i=1}^M\right)$. This process has three stages: neural message transmission (**S1**), neural message attention (**S2**) and neural message aggregation (**S3**).

In the *neural message transmission stage* (**S1**), each agent transmits its BEV-based feature map to the other agents. Since the BEV-based feature map summarizes the information of each agent, we consider it as a neural message. In the *neural message attention stage* (**S2**), each agent receives others' neural messages and determines the matrix-valued edge weights, which reflect the importance of the neural message from one agent to another at each individual cell. Since each agent has its unique pose, we leverage the collaboration graph to achieve feature transformation across agents. For the $i$th agent, the transformed BEV-based feature map from the $j$th agent is then $\mathbf{F}_{j \to i}^{\mathrm{s}} = \Gamma_{j \to i}(\mathbf{F}_i^{\mathrm{s}}) \in \mathbb{R}^{\bar{K} \times \bar{K} \times \bar{C}}$, where the transformation $\Gamma_{j \to i}(\cdot)$ is based on two ego poses $\boldsymbol{\xi}_j$ and $\boldsymbol{\xi}_i$. Now $\mathbf{F}_{j \to i}^{\mathrm{s}}$ and $\mathbf{F}_i^{\mathrm{s}}$ are supported in the same coordinate system. To determine the edge weights, we use the edge encoder $\Pi$ to correlate the ego feature map and the feature map from another agent; that is, the matrix-valued edge weight from the $j$th agent to the $i$th agent is $\mathbf{W}_{j \to i} = \Pi(\mathbf{F}_{j \to i}^{\mathrm{s}}, \mathbf{F}_i^{\mathrm{s}}) \in \mathbb{R}^{\bar{K} \times \bar{K}}$, where $\Pi$ concatenates two feature maps along the channel dimension and then uses four $1 \times 1$ convolutional layers to gradually reduce the number of channels from $2\bar{C}$ to $1$. Meanwhile, there is a softmax operation applied at each cell in the feature map to normalize the edge weights across multiple agents. Note that previous works [19, 20, 34] generally consider a scalar-valued edge weight to reflect the overall collaboration strength between two agents; while we consider a matrix-valued edge weight $\mathbf{W}_{j \to i}$, which models the collaboration strength from the $j$th agent to the $i$th agent with a $\bar{K} \times \bar{K}$ spatial resolution. In this matrix, each element corresponds to a cell in the BEV map, indicating a specific spatial region; thus, this matrix reflects the spatial attention at a cell-level resolution. In the *neural message aggregation stage* (**S3**), each agent aggregates the feature maps from all the agents based on the normalized matrix-valued edge weights. The updated feature map of the agent $i$ is $\mathbf{H}_i^{\mathrm{s}} = \sum_{j=1}^M \mathbf{W}_{j \to i} \odot \mathbf{F}_{j \to i}^{\mathrm{s}}$, where $\odot$ denotes the dot product with channel-wise broadcasting.

***Remark.*** The proposed collaboration graph is trainable because each matrix-valued edge weight is a trainable matrix to reflect the agent-to-agent attention in a cell-level spatial resolution; it is pose-aware, empowering all the agents to work with the synchronized coordinate system; furthermore, it is dynamic at each timestamp as each edge weight would adapt to the real-time neural messages. According to the proposed collaboration graph, the agents can discover the region requiring collaboration on the fly, and strategically select appropriate partners to request supplementary information.

**Decoder and header.** After collaboration, each agent decodes the updated BEV-based feature map. The decoded feature map is $\mathbf{M}_i^{\mathrm{s}} \leftarrow \Psi^{\mathrm{s}}(\mathbf{H}_i^{\mathrm{s}})$. To implement the decoder $\Psi^{\mathrm{s}}(\cdot)$, we progressively up-sample $\mathbf{H}_i^{\mathrm{s}}$ with four layers, where each layer first concatenates the previous feature map with the corresponding feature map in the encoder and then uses a $1 \times 1$ convolutional operation to halve the number of channels. Finally, we use an output header to generate the final detection outputs, $\widehat{\mathbf{Y}}_i^{\mathrm{s}} \leftarrow \Phi^{\mathrm{s}}(\mathbf{M}_i^{\mathrm{s}})$. To implement the header $\Psi^{\mathrm{s}}(\cdot)$, we use two branches of convolutional layers to classify the foreground-background categories and regress the bounding boxes.

## 2.2  Teacher: Early collaboration

During the training phase, an early-collaboration model, as the teacher, is introduced to guide the intermediate-collaboration model, which is a student. Similar to the student model, the teacher's pipeline has the feature encoder $\Theta^{\mathrm{t}}$, feature decoder $\Psi^{\mathrm{t}}$ and the output header $\Phi^{\mathrm{t}}$. Note that all the agents share the same teacher model to guide one student model; however, each agent provides the inputs with its own pose, and its inputs to both the teacher and student models should be well aligned.

---

[1]We consider a fully-connected bidirectional graph, and the weights for both directions are distinct.

**Feature encoder.** Let $\mathcal{X} = \mathcal{A}(\boldsymbol{\xi}_1 \circ \mathcal{X}_1, \boldsymbol{\xi}_2 \circ \mathcal{X}_2, ..., \boldsymbol{\xi}_M \circ \mathcal{X}_M)$ be a holistic-view 3D point cloud that aggregates all the points from all $M$ agents in the global coordinate system, where $\mathcal{A}(\cdot, ..., \cdot)$ is the aggregation operator of multiple 3D point clouds, and $\boldsymbol{\xi}_i$ and $\mathcal{X}_i$ are the pose and the 3D point cloud of the $i$th agent, respectively. To ensure the inputs to the teacher model and the student model are aligned, we transform the holistic-view point cloud $\mathcal{X}$ to an agent's own coordinate based on the pose information. Now, for the $i$th agent, the input to the teacher model $\boldsymbol{\xi}_i^{-1} \circ \mathcal{X}$ and the input to the student model $\mathcal{X}_i$ are in the same coordinate system. Similarly to the feature encoder in the student model, we convert the 3D point cloud $\boldsymbol{\xi}_i^{-1} \circ \mathcal{X}$ to a BEV map and use 2D convolutions to obtain the feature map of the $i$th agent in the teacher model, $\mathbf{H}_i^{\mathrm{t}} \in \mathbb{R}^{\bar{K} \times \bar{K} \times \bar{C}}$. Here we crop the BEV map to ensure it has the same spatial range and resolution with the BEV map in the student model.

**Decoder and header.** Similarly to the decoder and header in the student model, we adopt $\mathbf{M}_i^{\mathrm{t}} \leftarrow \Psi^{\mathrm{t}}(\mathbf{H}_i^{\mathrm{t}})$ to obtain the decoded BEV-based feature map and $\widehat{\mathbf{Y}}_i^{\mathrm{t}} \leftarrow \Phi^{\mathrm{t}}(\mathbf{M}_i^{\mathrm{t}})$ to obtain the predicted foreground-background categories and regressed bounding boxes.

**Teacher training scheme.** As in common teacher-student frameworks, we train the teacher model separately. We employ the binary cross-entropy loss to supervise foreground-background classification and the smooth $L_1$ loss to supervise the bounding-box regression. Overall, we minimize the loss function $\mathcal{L}^{\mathrm{t}} = \sum_{i=1}^M \mathcal{L}_{\mathrm{det}}(\mathbf{Y}_i^{\mathrm{t}}, \widehat{\mathbf{Y}}_i^{\mathrm{t}})$, where classification and regression losses are collectively denoted as $\mathcal{L}_{\mathrm{det}}$ and $\mathbf{Y}_i^{\mathrm{t}} = \mathbf{Y}_i^{\mathrm{s}}$ is the ground-truth detection in the perception region of the $i$th agent.

## 2.3 System training with knowledge distillation

Given a well-trained teacher model, we use both detection loss and knowledge distillation loss to supervise the training of the student model. We consider minimizing the following loss

$$\mathcal{L}^{\mathrm{s}} = \sum_{i=1}^M \left( \mathcal{L}_{\mathrm{det}}(\mathbf{Y}_i^{\mathrm{s}}, \widehat{\mathbf{Y}}_i^{\mathrm{s}}) + \lambda_{\mathrm{kd}} \mathcal{L}_{\mathrm{kd}}(\mathbf{H}_i^{\mathrm{s}}, \mathbf{H}_i^{\mathrm{t}}) + \lambda_{\mathrm{kd}} \mathcal{L}_{\mathrm{kd}}(\mathbf{M}_i^{\mathrm{s}}, \mathbf{M}_i^{\mathrm{t}}) \right).$$

The detection loss $\mathcal{L}_{\mathrm{det}}$ is similar to that of the teacher, including both foreground-background classification loss and the bounding box regression loss, pushing the detection result of each agent to be close to its local ground-truth. The second and third terms form a knowledge distillation loss, regularizing the student model to generate similar feature maps with the teacher model. The hyperparameter $\lambda_{\mathrm{kd}}$ controls the weight of the knowledge distillation loss $\mathcal{L}_{\mathrm{kd}}$ defined as follows

$$\mathcal{L}_{\mathrm{kd}}(\mathbf{H}_i^{\mathrm{s}}, \mathbf{H}_i^{\mathrm{t}}) = \sum_{n=1}^{\bar{K} \times \bar{K}} D_{\mathrm{KL}} \left( \sigma \left( (\mathbf{H}_i^{\mathrm{t}})_n \right) || \sigma ((\mathbf{H}_i^{\mathrm{s}})_n) \right),$$

where $D_{KL}(p(\mathbf{x})||q(\mathbf{x}))$ denotes the Kullback-Leibler (KL) divergence of distribution $q(\mathbf{x})$ from distribution $p(\mathbf{x})$, $\sigma(\cdot)$ indicates the softmax operation of the feature vector along the channel dimension, and $(\mathbf{H}_i^{\mathrm{s}})_n$ and $(\mathbf{H}_i^{\mathrm{t}})_n$ denote the feature vectors at the $n$th cell of the $i$th agent's feature map in the student model and teacher model, respectively. Similarly, the loss on decoded feature maps $\mathcal{L}_{\mathrm{kd}}(\mathbf{M}_i^{\mathrm{s}}, \mathbf{M}_i^{\mathrm{t}})$ can be introduced to enhance the regularization.

***Remark.*** As mentioned in Section 2.1, the feature maps output by the collaboration graph process in the student model is computed by $\{\mathbf{H}_i^{\mathrm{s}}\}_{i=1}^M \leftarrow \mathcal{M}_{\mathcal{G}_\Pi} \left( \{\mathbf{F}_i^{\mathrm{s}}\}_{i=1}^M \right)$. Intuitively, the feature map of the teacher model $\{\mathbf{H}_i^{\mathrm{t}}\}_{i=1}^M$ would be the desired output of the collaboration graph process $\mathcal{M}_{\mathcal{G}_\Pi}(\cdot)$. Therefore, we constrain all the post-collaboration feature maps in the student model to match the correspondences in the teacher model through knowledge distillation. This constraint would further regularize the upfront trainable components: i) the distilled student encoder $\Theta^{\mathrm{s}}$, which abstracts the features from raw measurements and produces the input to $\mathcal{M}_{\mathcal{G}_\Pi}(\cdot)$, and ii) the edge-weight encoder $\Pi$ in the distilled collaboration graph. Consequently, through knowledge distillation and back-propagation, the distilled student encoder would learn to abstract informative features from raw data for better collaboration; and the distilled edge-weight encoder would learn how to control the collaboration based on agents' features. In a word, our distilled collaboration network (DiscoNet) can comprehend feature abstraction and fusion via the proposed knowledge distillation framework.

# 3  Related Work

**Multi-agent communication.** As core components in a multi-agent system, communication strategy among agents has been actively studied in previous works [30, 7, 31]. For example, Comm-Net [32] adopted the averaging operation to achieve continuous communication in multi-agent system; VAIN [11] considered an attention mechanism to determine which agents would share information;

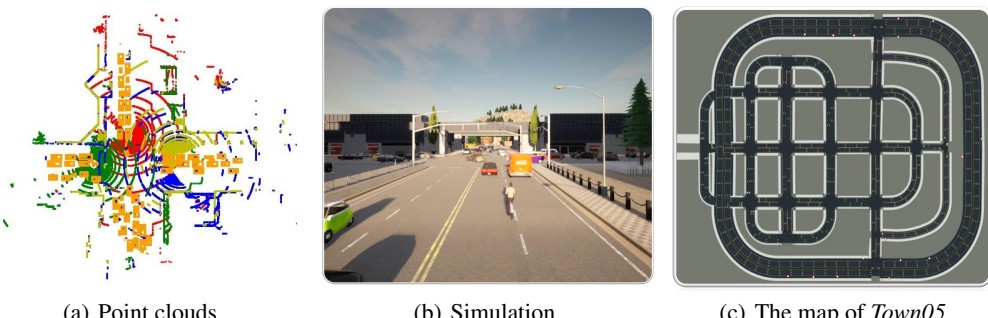

|   (a) Point clouds   |   (b) Simulation   |   (c) The map of *Town05*   |

Figure 3: V2X-Sim 1.0 dataset. (a) Point clouds from multiple agents in a top-down view (each color indicates an individual agent). (b) Snapshot of the rendered CARLA-SUMO co-simulation. (c) The map of *Town05* which is a squared-grid town with cross junctions and multiple lanes per direction.

ATOC [12] exploited an attention unit to determine what information to share with other agents; and TarMAC [5] implicitly learned a signature-based soft attention mechanism to let agents actively select which agents should receive the messages. In this work, we focus on the 3D perception task and propose a trainable, dynamic collaboration graph to control the communication among agents.

**Collaborative perception.** Collaborative perception is an application of multi-agent communication system to perception tasks [9, 36, 14]. Who2com [20] exploited a handshake communication mechanism to determine which two agents should communicate for image segmentation; When2com [19] introduced an asymmetric attention mechanism to decide when to communicate and how to create communication groups for image segmentation; V2VNet [34] proposed multiple rounds of message passing on a spatial-aware graph neural network for joint perception and prediction in autonomous driving; and [33] proposed a pose error regression module to learn to correct pose errors when the pose information from other agents is noisy.

**Knowledge distillation.** Knowledge distillation (KD) is a widely used technique to compress a larger teacher network to a smaller student network by pushing the student to approach the teacher in either output or intermediate feature space [10], and it has been applied to various tasks such as semantic segmentation [21], single-agent 3D detection [35], and object re-identification [13]. In this work, we apply the KD technique to a new application scenario: multi-agent graph learning. The teacher is an early-collaboration model with privileged information, and the student is an intermediate-collaboration model with limited viewpoint. The distilled collaboration graph could enable effective and efficient inference for the student model without teacher's supervision.

**Advantages and limitation of DiscoNet.** First, previous collaborative perception only rely on the final detection supervision; while DiscoNet leverages both detection supervision and intermediate feature supervision, acquiring more explicit guidance. Second, the collaboration attention in previous methods is a scalar, which cannot reflect the significance of each region; while DiscoNet uses a matrix, leading to more flexible collaboration at various spatial regions. Third, previous methods uses multiple-round collaboration: When2com needs at least one more round of communication after the first handshake, and V2VNet claims three rounds to ensure reliable performance; while DiscoNet only requires one round, suffering less from latency. Admittedly, DiscoNet has a limitation by assuming accurate pose for each agent, which could be improved by method like [33].

## 4 Experiment

### 4.1 V2X-Sim 1.0: A multi-agent 3D object detection dataset

Since there is no public dataset to support the research on multi-agent 3D object detection in self-driving scenarios, we build such a dataset named as V2X-Sim 1.0[2], based on both SUMO [15], a micro-traffic simulation, and CARLA [6], a widely-used open-source simulator for autonomous driving research. SUMO is firstly used to produce numerically-realistic traffic flow and CARLA is

---

[2]V2X (vehicle-to-everything) denotes the collaboration between a vehicle and other entities such as vehicle (V2V) and roadside infrastructure (V2I). V2X-Sim dataset is maintained on https://ai4ce.github.io/V2X-Sim/, and the first version of V2X-Sim used in this work includes the LiDAR-based V2V scenario.

Table 1: Detection comparison. Our method is in bold and * indicates the pose-aware version. The proposed DiscoNet is the best among intermediate collaboration. Even with 16 times compression, DiscoNet(16) still outperforms V2VNet without compression.

| Method | Collaboration Approach | | | Average Precision (AP) | |
|---|---|---|---|---|---|
| | Early | Intermediate | Late | IoU=0.5 | IoU=0.7 |
| Upper-bound | ✓ | ✗ | ✗ | 63.3 | 60.2 |
| | ✓ | ✗ | ✓ | 59.7 | 55.8 |
| When2com* [19] | ✗ | ✓ | ✗ | 45.7 | 41.7 |
| When2com [19] | ✗ | ✓ | ✗ | 45.7 | 41.8 |
| Who2com* [20] | ✗ | ✓ | ✗ | 44.3 | 40.3 |
| Who2com [20] | ✗ | ✓ | ✗ | 44.8 | 40.4 |
| V2VNet [34] | ✗ | ✓ | ✗ | 56.8 | 50.7 |
| **DiscoNet** | ✗ | ✓ | ✗ | **60.3** | **53.9** |
| **DiscoNet(16)** | ✗ | ✓ | ✗ | **58.5** | **53.0** |
| Lower-bound | ✗ | ✗ | ✓ | 57.6 | 54.2 |
| | ✗ | ✗ | ✗ | 45.8 | 42.3 |

Table 2: Detection performance by regularizing various layers via knowledge distillation (KD). $\mathbf{M}_i^s\{n\}(n = 1, 2, 3, 4)$ denotes different layers in the decoder. The performance significantly boots once the KD regularization is applied. Further regularization has a slight effect on the detection performance.

| KD Regularization | | | | | Average Precision (AP) | |
|---|---|---|---|---|---|---|
| $\mathbf{H}_i^s$ | $\mathbf{M}_i^s\{1\}$ | $\mathbf{M}_i^s\{2\}$ | $\mathbf{M}_i^s\{3\}$ | $\mathbf{M}_i^s\{4\}$ | IoU=0.5 | IoU=0.7 |
| ✗ | ✗ | ✗ | ✗ | ✗ | 57.2 | 52.3 |
| ✓ | ✗ | ✗ | ✗ | ✗ | 57.5 | 52.7 |
| ✓ | ✓ | ✗ | ✗ | ✗ | 58.2 | 52.7 |
| ✓ | ✓ | ✓ | ✗ | ✗ | 59.7 | 53.5 |
| ✓ | ✓ | ✓ | ✓ | ✗ | **60.3** | **53.9** |
| ✓ | ✓ | ✓ | ✓ | ✓ | 59.1 | 53.8 |
| ✗ | ✓ | ✓ | ✓ | ✓ | 58.3 | 52.6 |

employed to get realistic sensory streams including LiDAR and RGB data, from multiple vehicles located in the same geographical area. The simulated LiDAR is rotated at a frequency of 20Hz and has 32 channels; the range which denotes the maximum distance to capture is 70 meters. We use *Town05* with multiple lanes and cross junctions for dense traffic simulation; see Fig. 3.

V2X-Sim 1.0 follows the same storage format of nuScenes [2], a widely-used autonomous driving dataset. nuScenes collected real-world single-agent data; while we simulate multi-agent scenarios. Each scene includes a 20 second traffic flow at a certain intersection of *Town05*, and the LiDAR streams are recorded every 0.2 second, so each scene consists of 100 frames. We generate 100 scenes with a total of 10,000 frames, and in each scene, we randomly select 2-5 vehicles as the collaboration agents. We use 8,000/900/1,100 frames for training/validation/testing. Each frame has multiple samples, and there are 23,500 samples in the training set and 3,100 samples in the test set.

### 4.2 Quantitative evaluation

**Implementation and evaluation.** We crop the points located in the region of $[-32, 32] \times [-32, 32] \times [-3, 2]$ meters defined in the ego-vehicle $XYZ$ coordinate. We set the width/length of each voxel as 0.25 meter, and the height as 0.4 meter; therefore the BEV map input to the student/teacher encoder has a dimension of $256 \times 256 \times 13$. The intermediate feature output by the encoder has a dimension of $32 \times 32 \times 256$, and we use an autoencoder (AE) for compression to save bandwidth: the sent message is the embedding output by AE. After the agents receive the embedding, they will decode it to the original resolution $32 \times 32 \times 256$ for intermediate collaboration. The hyperparameter $\lambda_{\text{kd}}$ is set as $10^5$. We train all the models using NVIDIA GeForce RTX 3090 GPU. We employ the generic BEV detection evaluation metric: Average Precision (AP) at Intersection-over-Union (IoU) threshold of 0.5 and 0.7. We target the detection of the car category and report the results on the test set.

**Baselines and existing methods.** We consider the single-agent perception model; called *Lower-bound*, which consumes a single-view point cloud. The teacher model based on early collaboration with a holistic view is considered as the *Upper-bound*. Both lower-bound and upper-bound can involve late collaboration, which aggregates all the boxes and applies a global non-maximum suppression (NMS). We also consider three intermediate-collaboration methods, Who2com [20], When2com [19] and V2VNet [34]. Since the original Who2com and When2com do not consider pose information, we consider both pose-aware and pose-agnostic versions to achieve fair comparisons. All the methods use the same detection architecture and conduct collaboration at the same intermediate feature layer.

**Results and comparisons.** Table 1 shows the comparisons in terms of AP (@IoU = 0.5/0.7). We see that i) early collaboration (first row) achieves the best detection performance, and there is an obvious improvement over no collaboration (last row), *i.e.*, AP@0.5 and AP@0.7 are increased by 38.2% and 42.3% respectively, reflecting the significance of collaboration; ii) late collaboration improves the lower-bound (second last row), but hurts the upper-bound (second row), reflecting the unreliability of late collaboration. This is because the final, global NMS can remove a lot of noisy boxes for the lower-bound, but would also remove many useful boxes for the upper-bound; and iii) among the intermediate collaboration-based methods, the proposed DiscoNet achieves the best performance. Comparing to the pose-aware When2com, DiscoNet improves by 31.9% in AP@0.5 and 29.3% in

Table 3: Comparison of various intermediate collaboration strategies. KD indicates knowledge distillation. With KD and matrix-valued edge weights, our DiscoGraph outperforms the others.

| Collaboration Strategy | Scalar Weight | | | | Matrix Weight | | |
| --- | --- | --- | --- | --- | --- | --- | --- |
| | No | Sum | Average | Weighted Average | Max | Cat | DiscoGraph |
| w/o KD (AP@IoU 0.5/0.7) | 45.8/42.3 | 55.7/50.9 | 55.7/50.4 | 56.1/51.4 | 56.7/51.4 | 55.0/50.2 | **57.2/52.3** |
| w/ KD (AP@IoU 0.5/0.7) | 46.5/42.9 | 54.4/46.3 | 56.4/51.1 | 56.7/50.9 | 56.7/51.8 | 57.5/52.6 | **60.3/53.9** |
| Gain | 0.7/0.6 | -1.3/-4.6 | 0.7/0.7 | 0.6/-0.5 | 0.0/0.4 | 2.5/2.4 | **3.1/1.6** |
| Gain (%) | 1.5/1.4 | -2.3/-9.0 | 1.3/1.4 | 1.1/-1.0 | 0.0/0.7 | 3.3/4.6 | **5.4/3.1** |

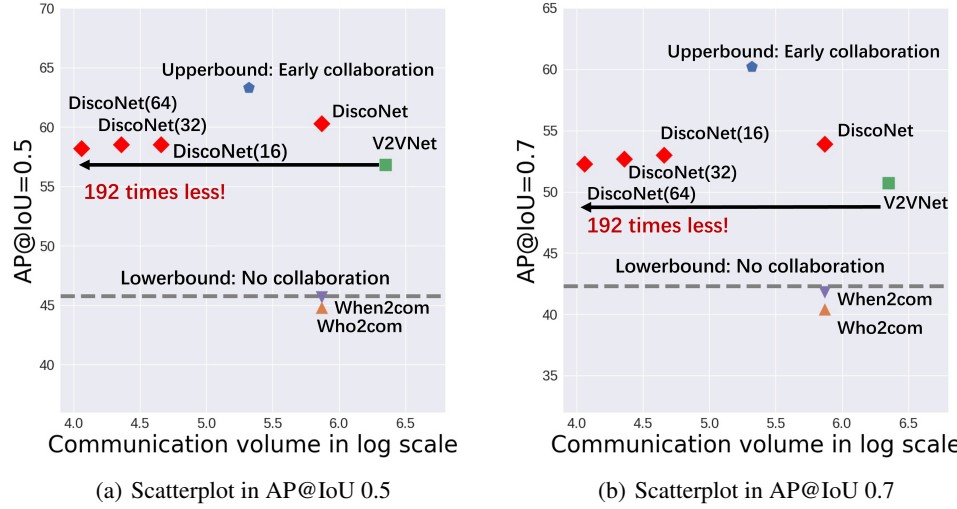

(a) Scatterplot in AP@IoU 0.5      (b) Scatterplot in AP@IoU 0.7

Figure 4: Performance-bandwidth trade-off. DiscoNet(64) achieves 192 times less communication volume and still outperforms V2VNet [34].

AP@0.7. Comparing to V2VNet, DiscoNet improves by 6.2 % in AP@0.5 and 6.3 % in AP@0.7. Even with 16-times compression by the autoencoder, DiscoNet(16) still outperforms V2VNet.

**Performance-bandwidth trade-off analysis.** Fig. 4 thoroughly compares the proposed DiscoNet with the baseline methods in terms of the trade-off between detection performance and communication bandwidth. In Plots (a) and (b), the lower-bound with no collaboration is shown as a dashed line since the communication volume is zero, and the other method is shown as one dot in the plot. We see that i) DiscoNet with feature-map compression by an autoencoder achieves a far-more superior trade-off than the other methods; ii) DiscoNet(64), which is DiscoNet with 64-times feature compression by autoencoder, achieves 192 times less communication volume and still outperforms V2VNet for both AP@0.5 and 0.7; and iii) feature compression does not significantly hurt the detection performance.

**Comparison of training times with and without KD.** Incorporating KD will increase the training time a little bit, from ~1200s to ~1500s per epoch (each epoch consists of 2,000 iterations). However, during inference, DiscoNet does not need the teacher model anymore and can work alone without extra computations. Note that our key advantage is to push the computation burden into the training stage in pursuit of efficient and effective inference which is crucial in the real-world deployment.

## 4.3 Qualitative evaluation

**Visualization of edge weight.** To understand the working mechanism of the proposed collaboration graph, we visualize the detection results and the corresponding edge weights; see Fig. 5 and Fig. 7. Note that the proposed edge weight is a matrix, reflecting the collaboration attention in a cell-level resolution, which is shown as a heat map. Fig. 5 (a), (b) and (c) show three exemplar detection results of Agent 1 based on the lower-bound, upper-bound and the proposed DiscoNet, respectively. We see that with the collaboration, DiscoNet is able to detect boxes in those occluded and long-range regions. To understand why, Fig. 5 (d) and (e) provide the corresponding ego edge weight and the edge weight from agent 2 to agent 1, respectively. We clearly see that with the proposed collaboration graph, Agent 1 is able to receive complementary information from the other agents. Take the first row in Fig. 5 as an example, we see that in Plot (d), the bottom-left spatial region has a much darker color, indicating that Agent 1 has less confidence about this region; while in Plot (e), the bottom-left spatial

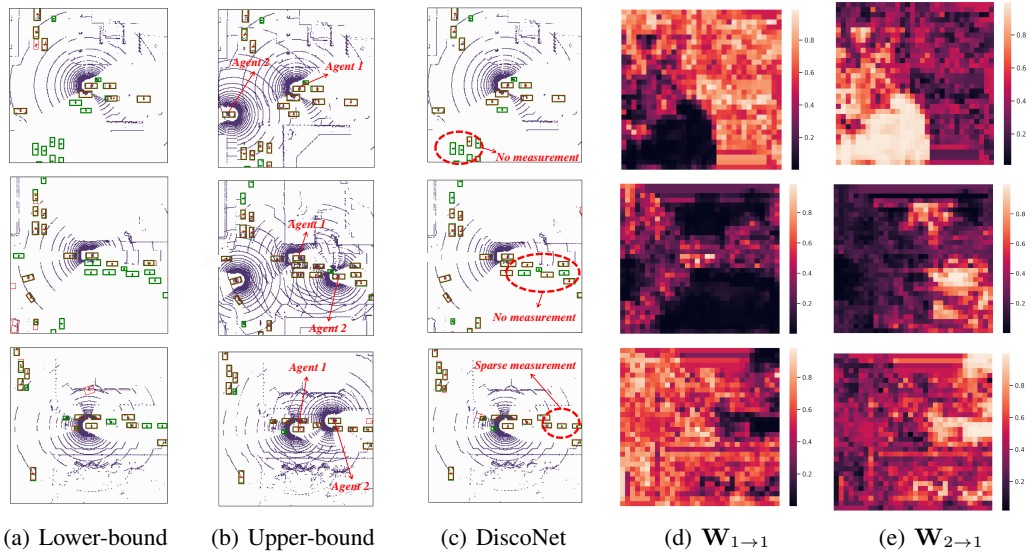

| (a) Lower-bound | (b) Upper-bound | (c) DiscoNet | (d) $\mathbf{W}_{1\rightarrow1}$ | (e) $\mathbf{W}_{2\rightarrow1}$ |

Figure 5: Detection and matrix-valued edge weight for Agent 1. Green and red boxes denote ground-truth (GT) and predictions, respectively. (a-c) shows the outputs of lower-bound, upper-bound and DiscoNet compared to GT. (d) Ego matrix-valued edge weight for Agent 1. Attention to the region with sparse/no measurement is suppressed. (e) Matrix-valued edge weight from Agent 2 to Agent 1. Attention to the region with complementary information from Agent 2 is enhanced.

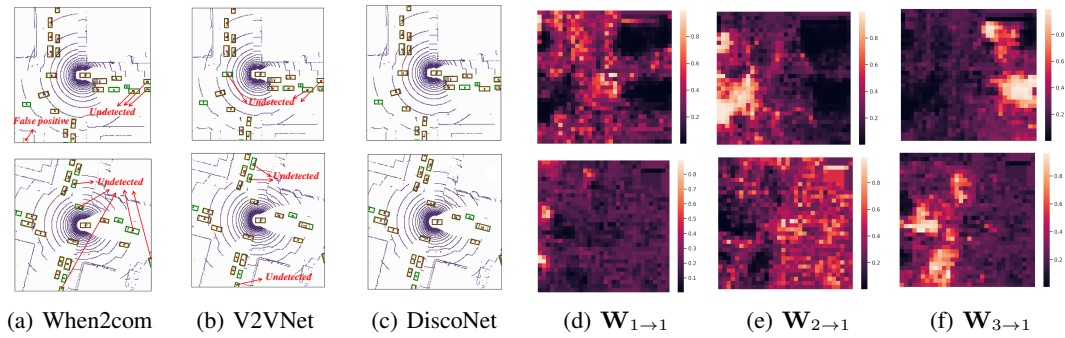

| (a) When2com | (b) V2VNet | (c) DiscoNet | (d) $\mathbf{W}_{1\rightarrow1}$ | (e) $\mathbf{W}_{2\rightarrow1}$ | (f) $\mathbf{W}_{3\rightarrow1}$ |

Figure 6: DiscoNet qualitatively outperforms the state-of-the-art methods. Green and red boxes denote ground-truth and detection, respectively. (a) Output of when2com. (b) Output of V2VNet. (c) Output of DiscoNet. (d)-(f) Matrix-valued edge weights. (Ego agent: 1; neighbour agents: 2 and 3.)

region has a much brighter color, indicating that Agent 1 has much stronger demands to request information from Agent 2.

**Comparison with When2com and V2VNet.** Fig. 6 shows two examples to compare the proposed DiscoNet with When2com, V2VNet. We see that DiscoNet is able to detect more objects. The reason is that both V2VNet and when2com employ a scalar to denote the agent-to-agent attention, which cannot distinguish which region is more informative; while DiscoNet can adaptively find beneficial region in a cell-level resolution; see the visualization of matrix-valued edge weights in Fig. 6 (d)-(f).

### 4.4 Ablation study

**Effect of collaboration strategy.** Table 3 compares the proposed method with five baselines using different intermediate collaboration strategies. *Sum* simply sums all the intermediate feature maps. *Average* calculates the mean of all the feature maps. *Max* selects the maximum value of all the feature maps at each cell to produce the final feature. *Cat* concatenates the mean of others' features with its own feature, and then uses a convolution layer to halve the channel number. *Weighted Average* appends one convolutional layer to our edge encoder $\Pi$, to generate a scalar value as the agent-wise edge weight. We use a matrix-valued edge weight to model the collaboration among agents. Without knowledge distillation, *Max* has achieved the second best performance probably because it ignores

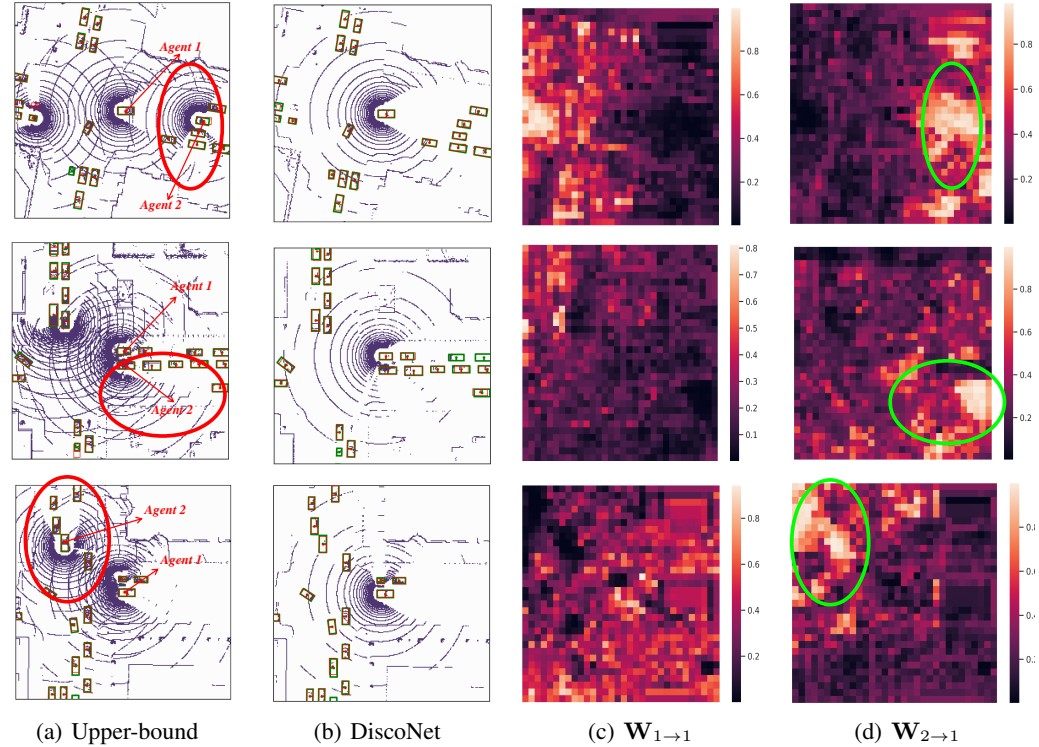

| (a) Upper-bound | (b) DiscoNet | (c) $\mathbf{W}_{1\to 1}$ | (d) $\mathbf{W}_{2\to 1}$ |

Figure 7: Detection and matrix-valued edge weight for Agent 1. Green/red boxes denote GT/predictions. (a)(b) show the outputs of upper-bound and DiscoNet. (c) Ego edge weight for Agent 1. (d) Edge weight from Agent 2 to Agent 1. The spatial regions containing complementary information for Agent 1 are highlighted by green circles in (d) and red circles in (a).

the noisy information with lower value. We see that with the proposed matrix-valued edge weight and knowledge distillation, DiscoGraph significantly outperforms all the other collaboration choices.

**Effect of knowledge distillation.** In Table 3 we investigate the versions with and without knowledge distillation, We see that i) for our method, the knowledge distillation can guide the learning of collaboration graph, and the agents can work collaboratively to approach the teacher's performance; ii) for *max* without a learnable module during collaboration, knowledge distillation has no impact; and iii) for *cat* and *average*, their performances are improved a little bit, as knowledge distillation can influence the feature abstraction process. Table 2 further shows the detection performances when we apply knowledge-distillation regularization at various network layers. We see that i) once we apply knowledge distillation to regularize the feature map, the proposed method starts to achieve improvement; and ii) applying regularization on four layers has the best performance.

## 5 Conclusion

We propose a novel intermediate-collaboration method, called distilled collaboration network (DiscoNet), for multi-agent perception. Its core component is a distilled collaboration graph (DiscoGraph), which is novel in both the training paradigm and the edge weight setting. DiscoGraph is also pose-aware and adaptive to perception measurements, allowing multiple agents with the shared DiscoNet to collaboratively approach the performance of the teacher model. To validate, we build V2X-Sim 1.0, a large-scale multi-agent 3D object detection dataset based on CARLA and SUMO. Comprehensive quantitative and qualitative experiments show that DiscoNet achieves appealing performance-bandwidth trade-off with a more straightforward design rationale.

**Acknowledgment.** This research is partially supported by the NSF CPS program under CMMI-1932187, the National Natural Science Foundation of China under Grant 6217010074, and the Science and Technology Commission of Shanghai Municipal under Grant 21511100900. The authors gratefully acknowledge the useful comments and suggestions from anonymous reviewers.

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
