# Supplementary Material: Learning Distilled Collaboration Graph for Multi-Agent Perception

**Yiming Li**
New York University
yimingli@nyu.edu

**Shunli Ren**
Shanghai Jiao Tong University
renshunli@sjtu.edu.cn

**Pengxiang Wu**
Rutgers University
pxiangwu@gmail.com

**Siheng Chen**[*]
Shanghai Jiao Tong University
sihengc@sjtu.edu.cn

**Chen Feng**[*]
New York University
cfeng@nyu.edu

**Wenjun Zhang**
Shanghai Jiao Tong University
zhangwenjun@sjtu.edu.cn

## I  Detailed information of the dataset

**Vehicle type and annotation.** Our dataset targets vehicle, bicycle and person detection in 3D point cloud, and we report the results of vehicle detection and leave the bicycle and person detection as the follow-up works. Noted that LiDAR point cloud would not capture car/person identity and thus 3D detection in point cloud does not involve privacy issue. There are twenty-one kinds of cars with various sizes and shapes in our simulated dataset, the names of vehicles in CARLA are listed below.

```
vehicle.bmw.grandtourer vehicle.bmw.isetta vehicle.chevrolet.impala
vehicle.nissan.patrol vehicle.tesla.cybertruck vehicle.tesla.model3
vehicle.mini.cooperst vehicle.volkswagen.t2 vehicle.toyota.prius
vehicle.citroen.c3 vehicle.dodge_charger.police vehicle.audi.tt
vehicle.mustang.mustang vehicle.nissan.micra vehicle.audi.a2
vehicle.jeep.wrangler_rubicon vehicle.carlamotors.carlacola
vehicle.audi.etron vehicle.mercedes-benz.coupe
vehicle.lincoln.mkz2017 vehicle.seat.leon
```

The 3D bounding boxes of different vehicles can be readily obtained without human annotations, and the LiDAR point cloud is aligned well with the camera image, as shown in Fig. I.

**CARLA-SUMO co-simulation.** We use CARLA-SUMO co-simulation for traffic flow simulation and data recording. Vehicles are spawned in CARLA via SUMO, and managed by the Traffic Manager. The script $spawn\_npc\_sumo.py$ provided by CARLA can automatically generate a SUMO network in a certain town, and can produce random routes and make the vehicles roam around, seen in Fig. II. Five hundred vehicles are spawned in $Town05$ and we record a log file with a length of five minutes, then we read out one hundred scenes from the log file at different intersections. Each scene includes a duration of twenty seconds, and there are totally $M(M = 2, 3, 4, 5)$ agents in a scene. Several examples of the generated scenes are shown in Fig. III.

**Dataset format.** We employ the dataset format of the nuScenes and extend it to multi-agent scenarios, seen in Fig. IV. Each log file can produce 100 scenes, and each scene includes 100 frames. Each frame covers multiple samples generated from multiple agents at the same timestamp. A sample includes the ego-pose of the agent, the sensor calibration information, and the corresponding annotations of its surrounding vehicles. Given a recorded log file, the dataset based on the log file can be generated

---

[*]Corresponding authors.

automatically with our tool, which does not require laborious manual annotations. Note that our dataset can be further enlarged to boost the object categories and traffic scenarios.

## II    Detailed architecture of the model

We use the main architecture of MotionNet [32] as our backbone, which uses an encoder-decoder architecture with skip connection. The input BEV map's dimension is $(c, w, h) = (13, 256, 256)$.

### II.1    Architecture of student/teacher encoder

We describe the architecture of the encoder below.

```
Sequential(
    Conv2d(13, 32, kernel_size=3, stride=1, padding=1)
    BatchNorm2d(32)
    ReLU()
    Conv2d(32, 32, kernel_size=3, stride=1, padding=1)
    BatchNorm2d(32)
    ReLU()
    Conv3D(64, 64, kernel_size=(1, 1, 1), stride=1, padding=(0, 0, 0))
    Conv3D(128, 128, kernel_size=(1, 1, 1), stride=1, padding=(0, 0, 0))
    Conv2d(32, 64, kernel_size=3, stride=2, padding=1)   ->(32,256,256)
    BatchNorm2d(64)
    ReLU()
    Conv2d(64, 64, kernel_size=3, stride=1, padding=1)
    BatchNorm2d(64)
    ReLU() ->(64,128,128)
    Conv2d(64, 128, kernel_size=3, stride=2, padding=1)
    BatchNorm2d(128)
    ReLU()
    Conv2d(128, 128, kernel_size=3, stride=1, padding=1)
    BatchNorm2d(128)
    ReLU() ->(128,64,64)
    Conv2d(128, 256, kernel_size=3, stride=2, padding=1)
    BatchNorm2d(256)
    ReLU()
    Conv2d(256, 256, kernel_size=3, stride=1, padding=1)
    BatchNorm2d(256)
    ReLU() ->(256,32,32)
    Conv2d(256, 512, kernel_size=3, stride=2, padding=1)
    BatchNorm2d(512)
    ReLU()
    Conv2d(512, 512, kernel_size=3, stride=1, padding=1)
    BatchNorm2d(512)
    ReLU() ->(512,16,16)
)
```

### II.2    Architecture of student/teacher decoder

The input of the decoder is the intermediate feature output by each layer of the encoder. Its architecture is shown below.

```
Sequential(
    Conv2d(512 + 256, 256, kernel_size=3, stride=1, padding=1)
    BatchNorm2d(256)
    ReLU()
    Conv2d(256, 256, kernel_size=3, stride=1, padding=1)
    BatchNorm2d(256)
    ReLU() ->(256,32,32)
```

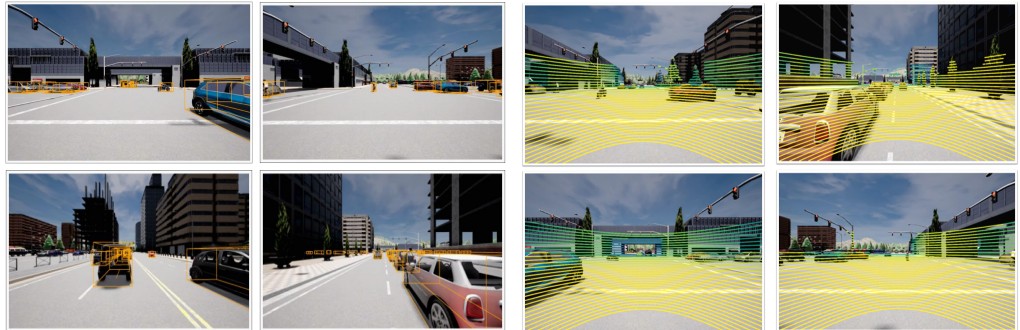

(a) Annotation of 3D bounding boxes      (b) Point cloud projected onto camera images

Figure I: Visualization of 3D bounding boxes annotation and the point cloud projected onto images.

```
    Conv2d(256 + 128, 128, kernel_size=3, stride=1, padding=1)
    BatchNorm2d(128)
    ReLU()
    Conv2d(128, 128, kernel_size=3, stride=1, padding=1)
    BatchNorm2d(128)
    ReLU() ->(128,64,64)
    Conv2d(128 + 64, 64, kernel_size=3, stride=1, padding=1)
    BatchNorm2d(64)
    ReLU()
    Conv2d(64, 64, kernel_size=3, stride=1, padding=1)
    BatchNorm2d(64)
    ReLU() ->(64,128,128)
    Conv2d(64 + 32, 32, kernel_size=3, stride=1, padding=1)
    BatchNorm2d(32)
    ReLU()
    Conv2d(32, 32, kernel_size=3, stride=1, padding=1)
    BatchNorm2d(32)
    ReLU() ->(32,256,256)
)
```

## II.3    Architecture of the edge encoder

The input whose dimension is $(c, w, h) = (512, 32, 32)$ of the edge encoder is the concatenation of the feature from ego agent and that from its neighbor agents. The output is a matrix-valued edge weight.

```
Sequential(
    Conv2d(512, 128, kernel_size=1, stride=1, padding=0)
    BatchNorm2d(128)
    ReLU()
    Conv2d(128, 32, kernel_size=1, stride=1, padding=0)
    BatchNorm2d(32)
    ReLU()
    Conv2d(32, 8, kernel_size=1, stride=1, padding=0)
    BatchNorm2d(8)
    ReLU()
    Conv2d(8, 1, kernel_size=1, stride=1, padding=0)
    BatchNorm2d(1)
    ReLU()
)
```

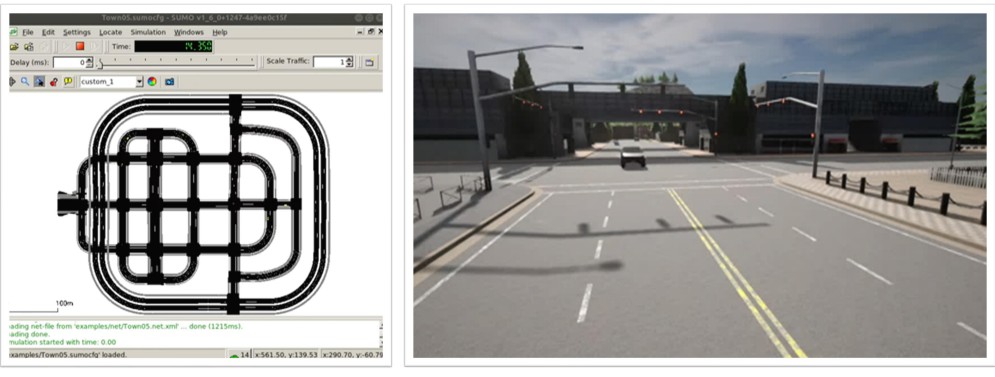

Figure II: CARLA-SUMO co-simulation.

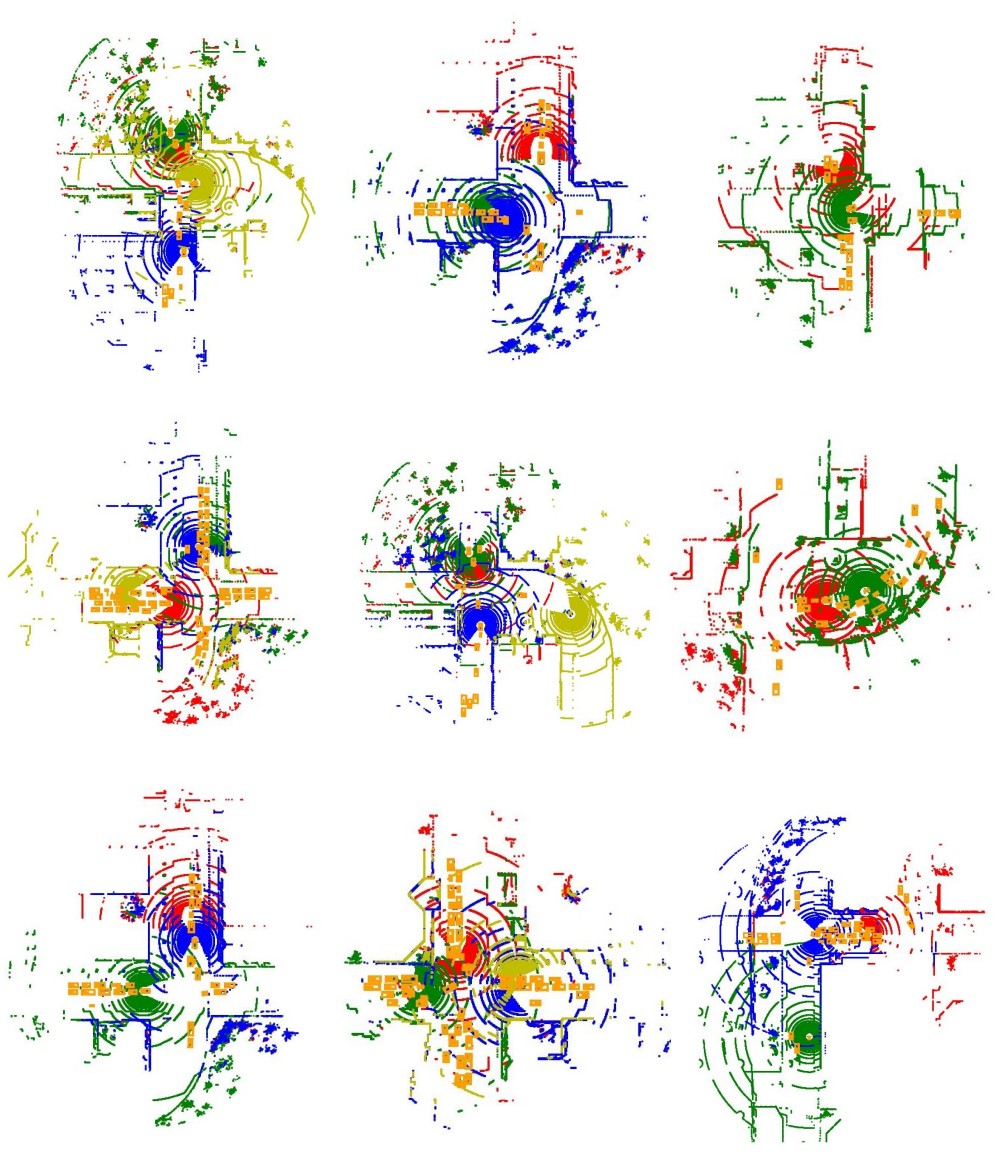

Figure III: Visualizations of the generated scenes in the bird's eye view. Each color represents an agent, and the orange boxes denote the vehicles in the scene.

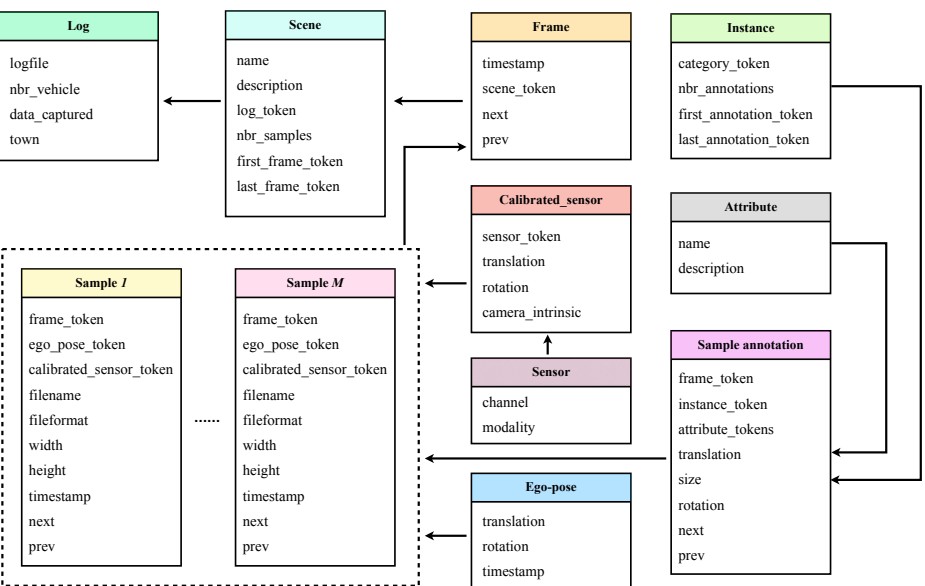

Figure IV: Schema of the dataset.