# OpenReview forum: "Learning Distilled Collaboration Graph for Multi-Agent Perception"
_NeurIPS.cc/2021/Conference — NeurIPS 2021 Poster_

### Official Review · Reviewer_rKjq · 2021-07-08

**Rating:** 7
**Confidence:** 4

**Summary:**

This paper tackles the problem of 3D detection from LiDAR in the multi-agent setting. It has two main contributions. The first contribution is the new method, which uses knowledge distillation from a teacher network using early (sensor) fusion to individual student networks that use intermediate (feature) fusion. Student networks fuse features from other student networks using a learnable spatial attention while being trained using both a standard detection loss as well as a regularization to be close to the teacher networks’ intermediate features. The method outperforms baselines in both AP and communication bandwidth
The second contribution is the release of the large scale synthetic dataset constructed with (SUMO and CARLA) used in the experiments.


**Limitations And Societal Impact:**

Perhaps authors could discuss security and safety implications of multiagent communication in general, and whether or not the specific proposed method has any differences in this regard compared to baselines.

**Main Review:**

The paper presents two nice contributions, 1) a novel method and 2) a benchmark dataset for multi-agent perception. However, I think some more in depth analysis is needed.

Regarding the dataset, can the authors comment on why the AP and IOU values seem to be quite lower than other comparable datasets? For example, V2V 93.1/89.9 AP at 0.5/0.7 IOU on the V2V dataset, but only achieves 57.0/49.1 on this dataset. The lower bound model (single agent perception) achieves only 45.8/40.6 which to my understanding is extremely low (e.g. compared to baselines for real world single agent settings, e.g. KITTI). This makes me more skeptical about where the gains DiscoGraph are actually coming from, and how the well baselines were tuned, etc. I believe such analysis is crucial for a paper which contributes a new dataset (which has the potential to be a widely used benchmark in the future!)

Regarding the method, one of the major sellings points is the decrease in communication bandwidth. However it seems like vanilla DiscoNet is not much better than the baselines, and it's largely the feature compression using 1x1 convolution that brings the gain. This makes me wonder how other baselines would perform using the same trick in feature compression. For example, it would be interesting to see if there is some aspect specific to the novelties in the Disco method that allows for maintaining AP with such aggressive compression, or if other baselines benefit from this easy trick as well*. I am also confused as to why the vanilla version of Disco actually has higher memory bandwidth than the ‘upper bound’ baseline. Finally, since it seems that DISCO maintains its AP with compression, I’d be interested in a simple baseline using some lidar compression (e.g. Draco as in V2V) for the ‘upper bound’ baseline.

In summary, I’d be more confident in the AP gains if the authors could shed light on why the numbers are so low across the board and share more details regarding tuning of baselines. I’d be more confident in the communication bandwidth gains if the feature compression trick was similarly applied to baselines.

Minor notes
Figure 2 caption
Since using Agent 1 as example, update notation F_i -> F_1
Give a sentence on where collaboration graph comes from
Line 129 - How is the original feature map F_i used to make H_i? I see in Figure 2 there is the self edge F_i->i. Maybe mention in text to be clear?
Section 2.3: What is the decoded feature map M? EDIT: I see it is mentioned later on in the paper, but I think it would be good to introduce it here first.
Figure 3: Why are RGB images included? Are they used at all in the method?
Figure 4: Units for x axis

*While the 1x1 convolution feature compression is simple and well known, as far as I know it has not been studied or used in the multi-agent perception setting, so I believe it is definitely a useful finding (due to how effective it is despite its simplicity as shown in this paper). However I am advocating to understand whether it is effective because of other things introduced in the Disco framework, or if it can be applied to many other scenarios. This of course affects how valuable the other architectural contributions are.

---

**EDIT:** I have updated my score, please see the response to the authors for details


**Time Spent Reviewing:**

3.5

---

> ### Author Response · Authors · 2021-08-10
> **Thanks for your constructive feedback**
>
> ### 1. Illustration of the detection performance on our dataset
>
> **1.1 Regarding the performance of V2VNet**
>
> V2VNet is developed by Uber ATG, a well-known company focusing on autonomous driving. However, neither its code nor its dataset is publicly available.  We thus have to re-implement V2VNet in PyTorch according to its pseudo-code. To make a fair comparison, we use the same backbone network for V2VNet and the proposed method, and evaluate on our dataset. Therefore, the experimental result on our dataset is of course different from that on the original V2V-Sim dataset. Indeed, the precision of V2VNet on V2V-Sim (93.1/89.9, AP@ 0.5/0.7) is extremely high, indicating that there is not much room for improvement on the V2V-Sim dataset. We thus design a much more challenging dataset for the community. *Please note that we will release all our resources including not only the dataset and code of DiscoNet but also our implementation of V2VNet, to facilitate the follow-up research for the community.*
>
> **1.2 Regarding the performance of our baseline detector**
>
> Our results are trustworthy and the relatively low APs stem from two aspects: the challenging dataset and the simple detector.
>
> * Regarding the dataset: We consider a challenging dataset to promote the necessity of collaborative perception. To validate this, we make a comparison with KITTI, a well-known autonomous driving dataset, from two aspects. First, we simulate a LiDAR with 32 channels and 70 meters perception range, while KITTI adopts Velodyne HDL-64E 3D laser scanner with 64 laser beams and 100 meters sensing range. Therefore, the point cloud in our dataset is much sparser. For example, across the entire dataset, the average numbers of 3D points falling into each vehicle for detection are 838 (KITTI) and 233 (ours), promoting the difficulty for 3D object detection. Second, there are much more vehicles to be detected in our dataset (\~23 vehicles per frame) compared to KITTI (\~7 vehicles per frame), and there are more occlusions in single-agent scenarios; see examples in Fig. 5(a). In summary, **our dataset is more challenging than KITTI due to sparser measurements and more occlusions**.
>
> * Regarding the detector: We use a classic anchor-based method as our baseline detector, which is mainly based on PointPillar. **Note that our main aim is to improve the performance via a well-designed collaboration strategy, instead of an advanced detector**. Therefore, we do not employ any complicated techniques that are generally used in single-agent detection, such as *data augmentation* and *temporal aggregation*, to avoid extra computations in our experiments. Moreover, all the methods are trained with the same setting to ensure that the performance gain comes from the collaboration instead of other techniques.
>
> To summarize, since our dataset is more challenging for 3D object detection, and we have not employed the techniques frequently used by the single-agent detector for saving cost, the lower bound achieves relatively low APs of 45.8/40.6. However, we have improved its performance with our DiscoNet significantly (60.2/54.0), and the only difference between the two methods is whether to collaborate for a fair comparison. We do agree with the reviewer that in-depth analysis is crucial for a new dataset. Therefore, **we will add more statistics analysis of our dataset, and we can provide multiple versions of the dataset to reflect various difficulty levels.**
> ***
>
> ### 2. Clarification of “compression”
>
> In general, compression means to represent the same data in a more compact form. Compression can be achieved from multiple perspectives. For example, **dimensionality reduction** reduces the size of data volume by approximating data in a lower-dimensional coordinate system, achieving the goal of compression; and **source coding/entropy coding** reduces data redundancy by exploring data’s statistical property, also promoting compression. The output of dimensionality reduction is lower-dimensional features and the output of source coding is a sequence of binary codes. Note that in V2VNet, the LiDAR compression tool, DRACO, considers source coding; while in our work, the 1x1 convolution considers dimensionality reduction. **Since all the methods do not consider source coding in our comparison, we achieve a fair comparison.**
>
> As the reviewer suggested, we can add the source coding in our experiments. Here we consider lossless source coding for both point cloud and the intermediate feature map after the existing dimensionality reduction. We randomly select five point cloud _.bin_ files with an average size of 870KB, and DRACO can be utilized to compress them into .bin files with an average size of 390KB. The corresponding intermediate feature maps with an original resolution of 32$\times$32$\times$256 can be reduced into a resolution of 32$\times$32$\times$4 with our dimensionality reduction, and the memory size is reduced from 1MB to 16KB on average, and we further use the standard source coding tool, 7zip, to compress the feature map to .zip files with an average size of 10.3KB. **This reflects that even with lossless source coding, the compression ratio of an intermediate feature map (1MB->10.3KB) is still much larger than that of a 3D point cloud (870KB->390KB).**
>
> Regarding the lossy compression, we use the default configuration of draco to compress the point cloud (compression level is 7 and quantization parameter is 11), and the significant compression ratio **(870KB -> 23.9KB, \~36 times)** will hurt the performance largely **(64.2/60.3 -> 55.6/47.9)**. In contrast, our method can still achieve satisfactory performance **(60.2/54.0 -> 58.4/52.0)** even with a large compression ratio **(1MB->10.3KB, \~100 times)**. Note that in the compression community, it is well known that the 3D point cloud is more difficult to compress because of its irregular structure compared to the traditional image data (our feature map could be considered as a pseudo-image) [1,2].
>
> Regarding the communication volume, we adopt the broadcast mode, e.g., in sensor fusion, there are averagely 24,363 points per vehicle and 2.8 vehicles per scene, then the average communication volume in each scene is log(3$\times$24,363$\times$2.8)=5.3, where 3 denotes xyz axis; in original DiscoNet, we transmit the feature map with the size of 32$\times$32$\times$256 (W$\times$H$\times$C), then the average communication volume is log(32$\times$32$\times$256$\times$2.8)=5.9; in DiscoNet(64) with dimensionality reduction, the average communication volume is log(32$\times$32$\times$4$\times$2.8)=4.1.
>
> [1] Zhang, Cha, D. Florêncio and Charles T. Loop. “Point cloud attribute compression with graph transform.” ICIP (2014).
>
> [2] Wang, Jianqiang, Hao Zhu, Haojie Liu and Zhan Ma. “Lossy Point Cloud Geometry Compression via End-to-End Learning.” IEEE Transactions on Circuits and Systems for Video Technology (2021).
> ***
> ### 3. Dimensionality reduction on other baselines
> As the reviewer suggested, we conduct experiments that incorporate the dimensionality reduction into the V2VNet and When2com. Our experiments validate that with the proposed 1x1 convolution, both V2VNet and When2com could still achieve comparable performance even when their channel dimensions are significantly reduced. The results are shown below.
>
> | Compression ratio | 1 | 16 |32 | 64 |
> | :-----| ----: | :----: |:----: |:----: |
> | DiscoNet | 60.2/54.0|59.0/53.8 |58.8/53.7|58.4/52.0 |
> |V2VNet |57.0/49.1 | 58.6/52.1 |57.8/51.9 | 58.1/51.7 |
> |When2com | 47.9/42.9 | 46.4/42.3 |45.1/41.3 | 45.2/40.0 |
>
> We see that 1) the proposed dimensionality reduction module can be generally employed in the intermediate collaboration methods, and can marginally influence the performance; 2) DiscoNet consistently outperforms V2VNet across all the compression ratios; 3) given limited communication bandwidth, the performances of DiscoNet and V2VNet become similar. This is not surprising because when the communication bandwidth is zero, all these methods would share the similar performance, which is the lower bound. Therefore, for low bandwidth, slight AP gain is still very valuable; and 4) for V2VNet, the dimensionality reduction can even improve the performance a bit, possibly because the optimization of the original V2VNet is hard and some features are redundant and ineffective. With the dimensionality reduction by our 1x1 convolution, the features could be refined. *Moreover, we emphasize that the V2VNet’s communication volume is always **three times** as much as DiscoNet since it consumes **three rounds** of message passing. This can also introduce **extra latency** in practice, damaging the communication efficiency.*
>
> Admittedly, 1x1 convolution reduces the channel dimension and brings significantly cheaper communication bandwidth; **we would like to emphasize our technical contributions on knowledge-distillation-based framework and matrix-value edge weights, which not only enable the proposed DiscoNet to achieve a better performance-bandwidth tradeoff than V2VNet and When2com at various multiple communication volumes, but also bring more straightforward design rationale.**
> ***
> ### 4. Minor comments
> We thank you sincerely for your meticulous evaluation, and we will carefully revise our paper based on your comments. For the safety implications, based on the estimation of the U.S. NHTSA, there would be a minimum of **13%** reduction in traffic accidents if a V2V system were implemented, which means **439,000** fewer crashes every year. Therefore, multi-agent perception could fundamentally promote the safety of autonomous driving by providing more reliable perception results. Admittedly, **adversarial attacks** may also emerge since the communication channel could be maliciously manipulated by the adversarial agents, and **data privacy** issues during communication are also worth studying. We thank you very much and will add the suggested discussion.

---

> > ### Comment · Reviewer_rKjq · 2021-08-20
> > **Thanks to the authors for their detailed reply**
> >
> > ## Response to the authors
> > > 1. Illustration of the detection performance on our dataset
> >
> > Thank you for the explanation and the analysis on the dataset, this addresses my concerns and I believe is very valuable.
> >
> > >  2.  Clarification of “compression”
> >
> > Pardon my misuse of compression vs dimensionality reduction. I appreciate the new experiments and I think its valuable to see that feature compression is outperfoming standard lidar compression
> >
> > > 3. Dimensionality reduction on other baselines
> >
> > Thank you for running the experiment that I suggestioned. It seems that these results do confirm my concern that a large component of the DiscoNet bandwidth reduction gains come from this simple trick, rather than the difference in architecture (that is, other architectures equally can reduce bandwidth by 64x without much performance loss). While the authors emphasize their technical contribution is the DiscoNet architecture, the AP gains when put in context are a bit less exciting when it seems that the architecture itself is almost orthogonal to the main feature compression technique.
> >
> > ---
> >
> > ## Final comments
> > With that being said, I am willing to raise my score from a 5 to a 7 given that the new anlaysis and experiments improve the paper. I believe the dataset + analysis and the open source baseline implementations will be useful for the community. However, I'm still not 100% convinced on the benefit of the DiscoNet architecture itself, as the AP gains at 64x compression are only marginally better than V2V. Yes it is true that DiscoNet still requires 3 times less communication than V2V, but 192 -> 3x is a big drop. Perhaps the authors can focus more on the dataset contribution instead of the architecture.

---

> > > ### Author Response · Authors · 2021-08-21
> > > **We really appreciate your positive and valuable feedback**
> > >
> > >  We thank you sincerely for your approval of our dataset contribution and your endorsement of our paper. We will release our dataset with in-depth analysis as soon as possible based on your kind suggestions. Meanwhile, we would like to make a **technical comparison** between the proposed **DiscoNet** and the previous **V2VNet** and further emphasize the superiority of the proposed method in terms of latency.
> > >
> > > ### 1. Technical Comparison
> > >
> > > * **(i)** In terms of the **training** of collaboration strategy, V2VNet purely relies on the end-to-end learning and the supervision from final detection labels, which lacks of interpretability, while the proposed DiscoNet proposes a novel teacher-student architecture, which provides an explicit collaboration objective; that is, when multiple single-view students collaborate together, they should act like the holistic-view teacher.
> > >
> > > * **(ii)** To model pairwise collaboration between agents, both V2VNet and the proposed DiscoNet adopt pose-aware graphs; however, the underlying **edge weights** are distinct. V2VNet considers a scalar-value edge weight, which can only reflect the attention from one agent to another as a whole; while the proposed DiscoNet proposes a matrix-valued edge weight, which can reflect the attention from one agent to another at each spatial location.
> > >
> > > * **(iii)** To propagate agents’ information, V2VNet adopts multiple-round communication, while the proposed DiscoNet only needs a single-round communication, which would significantly reduce the notorious **latency issue**.
> > >
> > > * **(iv)** To aggregate other agents’ information, V2VNet employs a **sophisticated** convolutional gated recurrent unit (ConvGRU)-based module, while the proposed DiscoNet uses a **straightforward and lightweight** graph convolution layer.
> > >
> > > *In summary, the technical novelties of the proposed DiscoNet should not be ignored. Even though its detection performance does not beat V2VNet by an extremely large margin, DiscoNet with a straightly novel architecture consistently outperforms V2VNet in terms of the bandwidth-detection tradeoff.*
> > >
> > > ### 2. Latency
> > > As mentioned in the third point in Technical comparison, V2VNet needs **multiple-round** communication and the proposed DiscoNet only needs **once**. Therefore, DiscoNet would significantly reduce the latency, which is one of the most critical issues in communication. Actually, the **latency issue** also attracts the attention in the perception community [1,2,3]. For example, the study of Uber ATG [2] specifically points out that  *“Modern perception algorithms wait for the full sweep to be built before processing the data, which introduces an additional latency of up to 100ms. As a consequence, by the time an output is produced, it no longer accurately reflects the state of the world. This poses a big challenge, as robotics applications **require minimal reaction times**, such that maneuvers can be **quickly planned** in the event of a safety-critical situation.”* In the task of collaborative perception, communication is necessary and critical. Even though resolving the latency issue is not the main topic of this paper, we still consider this is one of the advantages of the proposed method.
> > >
> > >
> > > [1] Li, Mengtian, Yu-Xiong Wang, and Deva Ramanan. "Towards streaming perception." In European Conference on Computer Vision, pp. 473-488. Springer, Cham, 2020.
> > >
> > > [2] Frossard, Davi, Simon Suo, Sergio Casas, James Tu, Rui Hu, and Raquel Urtasun. "Strobe: Streaming object detection from lidar packets." In Conference on Robot Learning, 2021.
> > >
> > > [3] Han, Wei, Zhengdong Zhang, Benjamin Caine, Brandon Yang, Christoph Sprunk, Ouais Alsharif, Jiquan Ngiam, Vijay Vasudevan, Jonathon Shlens, and Zhifeng Chen. "Streaming object detection for 3-d point clouds." In European Conference on Computer Vision, pp. 423-441. Springer, Cham, 2020.

---

### Official Review · Reviewer_4Gjd · 2021-07-16

**Rating:** 6
**Confidence:** 4

**Summary:**

This paper proposes a learning algorithm for multi-agent perception tasks using ideas from knowledge distillation ideas. They model the relationship between each agent as a collaboration graph to be learned and use richer data in the teacher model to distill its knowledge to the student model. Through extensive analysis, they have shown that the approach can outperform other baseline methods in the multi-agent and single-agent perception tasks of object detection. They also combined two datasets, in order to introduce their multi-agent dataset for object detection.

**Limitations And Societal Impact:**

It does not apply to this paper, nor do they discussed this in the paper.

**Main Review:**

In this paper, the authors proposed a new learning algorithm for multi-agent perception tasks, which is interesting in its own right. The proposed task is similar to the combination of ideas from privileged information with knowledge distillation for this specific application. The teacher has access to more information while the student model process single-view data and aggregate them using a collaboration graph. The usage of the knowledge distillation in this task is nontrivial and novel, which seems to be useful in practice. The organization of the paper is clear and the related baselines are discussed and compared with. I have some concerns regarding this paper:

 - How the structure of the collaboration graph is decided? is it a full graph or there might be some links missing? Is the graph bidirectional or directed? If it is bidirectional, are the weights for both directions shared or separate? I think the description of this graph is missing from the text.

- In Figure 7.d the weights of the link between agent 2 and 1 is presented, when there is no teacher. This is confusing since even without the teacher, these parameters should be learned using the loss on different views. Then, how is that this matrix is converging to a single value?

- Since this algorithm has some overhead computation during the training, it is beneficial to compare training times with and without KD as well.

**Time Spent Reviewing:**

4

---

> ### Author Response · Authors · 2021-08-10
> **Thanks for your constructive feedback**
>
> **1. The structure of the collaboration graph**
>
> We thank our reviewer for pointing that out. Our collaboration graph is fully-connected and bidirectional, and the weights for both directions are distinct.
>
> **2. The degeneration of matrix-valued edge weight in Figure 7.d**
>
> This is indeed a surprising and interesting fact. We have confirmed that the matrix-valued edge weight gradually converge to a constant value since the first few epochs. There might be two reasons. First, there is a softmax operation applied at each pixel in the BEV map to normalize the edge weights across multiple agents. Therefore, for the same agent, the original edge weights at various pixels could be different, but after normalization, all the edge weights are the same. Second, without intermediate supervision, the edge encoder is not able to learn which spatial area could be more informative, resulting in a constant-valued edge weight, which is the easiest one for optimization. This reflects the significance of the proposed KD-based intermediate supervision.
>
> **3. Comparison of training times with and without KD**
>
> Incorporating knowledge distillation will increase the training time a little bit, from ~1200s to ~1500s per epoch on a NVIDIA GeForce RTX 3090 GPU  (each epoch consists of 2,000 iterations). However, during inference, DiscoNet does not need the teacher model anymore and can work alone without extra computations. Note that our key advantage is to push the computation burden into the training stage in pursuit of efficient and effective inference which is crucial in the real-world deployment. We will add the comparison of training time with and without KD, and we really appreciate your constructive suggestions.

---

> > ### Comment · Reviewer_4Gjd · 2021-08-22
> > **Insights regarding the graph**
> >
> > I appreciate the authors' effort in addressing my comments. I still have some suggestions regarding the graph weights. In the paper you have visualized the weights between different agents, however, I think it would be beneficial to further discuss the intuition behind the learned weights. How they reveal the importance of different collaborations for a specific task. At this stage, I am satisfied with the answers and will keep my score.

---

> > > ### Author Response · Authors · 2021-08-22
> > > **Thanks for your constructive comments**
> > >
> > > ### 1. Design rationale of matrix-valued edge weight.
> > >  In the proposed collaboration graph, each edge models the collaboration between two agents and the corresponding edge weight reflects the attention level of such a pairwise collaboration. *Since each agent has its own spatial visibility due to occlusion or long-range issues, it has different levels of collaboration demands at various spatial regions.* To promote the collaboration at a **finer spatial resolution**, we propose a novel **matrix-valued edge weight** that reflects the levels of collaboration demands at various **spatial regions**. The entire matrix models a bird’s-eye-view spatial region and each element in this matrix models a specific spatial region. For example, given an edge that models the collaboration from Agent 2 to Agent 1, when the $(i,j)$th element of this edge weight has a higher value, this means that Agent 1 has stronger demands to receive information about the $(i,j)$th spatial region from Agent 2. In comparison, previous works consider a **scalar-value edge weight**, which has a very **coarse spatial resolution**. *Therefore, the previous method considers that no matter what the visibility of Agent 2 is, Agent 1 always has all the same collaboration demands from Agent 2 across any spatial region, while the proposed method is able to distinguish the information at which spatial region from Agent 2 is more informative to Agent 1.*
> > >
> > > ### 2. Implementation of matrix-valued edge weight
> > >  With the ego feature and the received features from the other agents, we concatenate them ﻿along the channel dimension and feed into an edge-weight-computation network with multiple 1x1 convolutional layers to ﻿gradually reduce the number of channels to 1, followed by a softmax function over all agents; and then we get the edge weight of collaboration graph (a mask as shown in Figure 5). These edge weights are trained based on both the guidance of the **teacher network** and the final **detection supervision**.
> > >
> > > ### 3. Visualization of matrix-valued edge weight
> > > For example, Figure 5 (e) has visualized the matrix-valued edge weight that corresponds to the edge from Agent 2 to Agent 1. *At any spatial region, a brighter color means a stronger demand of Agent 1 to collaborate with Agent 2.* Specifically, in the first row of Figure 5, all five plots are aligned to share the same bird’s-eye-view spatial region. Based on Plot (a), we see that the bottom-left spatial region is invisible to Agent 1 due to occlusion, so we suppose that Agent 1 should be unsure about this region and have higher demands to request information from Agent 2 about this region. To validate that the proposed matrix-valued edge weight can capture this, we see that in Plot (d), the bottom-left spatial region has a much darker color, indicating that Agent 1 has less confidence about this region; while in Plot (e), the bottom-left spatial region has a much brighter color, indicating that Agent 1 has much stronger demands to request information from Agent 2.
> > >
> > > In summary, according to the proposed collaboration graph with the matrix-valued edge weights, the agents can discover the spatial regions with various collaboration demands on the fly, and strategically select appropriate partners to request supplemental information. We thank your very much and we will add more discussions around the proposed graph based on your suggestions.

---

### Official Review · Reviewer_EHoG · 2021-07-16

**Rating:** 7
**Confidence:** 3

**Summary:**

This paper proposed a distilled collaboration graph (DiscoGraph) to promote better performance-bandwidth trade-off for multi-agent perception. The contribution lies in two folds:  (1) the paper raised a collaborative teacher-student framework to train DiscoGraph, (2)  Matrix-valued edge weight is introduced to DiscoGraph to allow each agent to adaptively highlight the informative regions.


**Ethical Concerns:**

Not any

**Ethics Review Area:**

["I don’t know"]

**Limitations And Societal Impact:**

Add discussion of differences between their proposed method and the SOTA distillation method.

**Main Review:**

Pros:
1. In general, the paper is well organized and easy to follow.
2. The proposed intermediate-collaboration method sounds somewhat interesting and makes sense.
3. The figures and tables in this paper are well designed. All figures are not only informative but also beautiful.
4. Sufficient experiments on several datasets are conducted to verify the effectiveness of the proposed method.

Cons:
The related works are not well discussed in this paper. As distillation has become a hot area in recent years, there are extensive new methods investigating its underlying power. However, the authors utilized the distillation method without analyzing the relevance of previous methods.


**Time Spent Reviewing:**

1

---

> ### Author Response · Authors · 2021-08-10
> **Thanks for your valuable feedback**
>
> ### Relations to distillation methods
>
> The comments from Reviewer 4Gjd have clearly summarized our novelty in distillation. "The proposed task is similar to the combination of ideas from privileged information with knowledge distillation for this specific application. The teacher has access to more information while the student model process single-view data and aggregate them using a collaboration graph. The usage of the knowledge distillation in this task is nontrivial and novel, which seems to be useful in practice."
>
> Actually, we had a paragraph in the related works to review existing distillation methods, however, due to the space reason, we removed this paragraph. Once given more space in the camera ready version, we will include it again. Here we post our discussion on distillation.
>
> "Knowledge distillation (KD) is a widely-used technique to compress a large teacher network to a small student network. KD is first introduced in [1], where a student network is regularized to approximate the teacher network in the output space. Afterwards, [2] further allows a student network to approach the teacher network in the intermediate feature space. Along this research line, KD has been modified by exploring more kinds of dark knowledge [3,4]. Furthermore, KD has been applied to numerous computation-expensive visual tasks, e.g., semantic segmentation [5], object detection [6], and face recognition [7]. In this work, we apply the KD techniques to collaborative perception, where the teacher is a 3D detector taking a multi-view 3D point cloud as input, and the student is another 3D detector whose input is a single-view 3D point cloud. KD can guide the optimization of collaboration among multiple agents, and help the agents which share the student model collaboratively approach the performance of the teacher model."
>
> [1] Hinton, Geoffrey E., Oriol Vinyals and J. Dean. “Distilling the Knowledge in a Neural Network.” ArXiv abs/1503.02531 (2015).
>
> [2] Romero, Adriana, Nicolas Ballas, Samira Ebrahimi Kahou, Antoine Chassang, Carlo Gatta, and Yoshua Bengio. "Fitnets: Hints for thin deep nets." arXiv preprint arXiv:1412.6550 (2014).
>
> [3] Zagoruyko, Sergey and N. Komodakis. “Paying More Attention to Attention: Improving the Performance of Convolutional Neural Networks via Attention Transfer.” ICLR (2017).
>
> [4] Yim, Junho, Donggyu Joo, Ji-Hoon Bae and Junmo Kim. “A Gift from Knowledge Distillation: Fast Optimization, Network Minimization and Transfer Learning.” CVPR (2017).
>
> [5] Xie, J., Bing Shuai, Jianfang Hu, Jing-Jhih Lin and W. Zheng. “Improving Fast Segmentation With Teacher-Student Learning.” BMVC (2018).
>
> [6] Wang, Tao, L. Yuan, Xiaopeng Zhang and Jiashi Feng. “Distilling Object Detectors With Fine-Grained Feature Imitation.” CVPR (2019).
>
> [7] Ge, Shiming, Shengwei Zhao, Chenyu Li and Jia Li. “Low-Resolution Face Recognition in the Wild via Selective Knowledge Distillation.” IEEE Transactions on Image Processing 28 (2019).

---

### Official Review · Reviewer_AgpJ · 2021-07-19

**Rating:** 5
**Confidence:** 4

**Summary:**

The submission proposes a new multi-agent communication algorithm to simultaneously maximize what information is shared while trying to reduce how much data is transferred between agents. It calls this algorithm DiscoGraph and showcases its utility for a 3D object detection task built on top of CARLA and SUMO.

There are two contributions within the DiscoGraph algorithm. First, to maximize information sharing between agents, they use a teacher-student framework. The teacher is only used during training and discarded during inference. The teacher receives raw sensory LIDAR data and aggregates all the data across all the agents. It then uses this combined data to do 3D object detection. The features generated by the teacher therefore contain information seen by all the agents. DiscoGraph distills this knowledge to the individual agents by ensuring that the features generated by the student networks match the one generated by the teacher. The student network for a single agent does not get to see the raw sensory data from all other agents. Instead, it encodes the features into a lower dimensional representation and shares that information through a graph-attention network formulation. The resulting features, if the graph-attention network is trained well, encodes relevant information from the features encoded by the other agents. This is enforced using a KL loss with the teacher network’s representations. In other words, the graph-attention network must learn to share appropriate information such that the features encoded by each agent’s student networks is the same as the features encoded by the teacher that has all the raw data.

The second contribution within the DiscoGraph algorithm is the per spatial location attention generated within the graph-attention network. This allowed the graph updates to attend over specific spatial locations when sharing information.

The paper ablates these two contributions and shows that both are in fact useful for the main task of 3D object detection.


**Limitations And Societal Impact:**

The submission could do a better job articulating the societal impacts of this work. It is clear to me that being able to communicate effectively about what each agent sees is a valuable improvement to multi-agent performance. But it’s not clear to me exactly what kinds of tasks (other than multi-agent 3D object detection) would benefit from such effective communication.

It’s also hard to understand the limitations of the method. The dataset appears to be limited to a particular town. There needs to be a longer discussion around the dataset -- what kinds of scenarios does it test for? Are there common types of errors the agents make? How consistent are the agents when they share an observation space? Are there situations that are not evaluated by the dataset?


**Main Review:**


I actually like the method, the experimental setup, the ablations, and the results are compelling. However, there are two key problems that have prevented me from giving this submission a higher score. First, the paper could be written more clearly. Second, the paper doesn’t properly explain why it develops a new benchmark/task and doesn’t use existing ones proposed by the baselines [17, 18, 31].

**Writing: The abstract and introduction dives into details about the model before even explaining why we need such a model. 3D multi-agent object detection isn’t as established a problem space as object recognition; there needs to be more text explaining the motivation, ecological validity and utility of such a task. This is especially important since the submission introduces a new benchmark. What design options were considered when creating the dataset? For example, what does it mean to perform at XX average precision? What are the realistic communication bandwidth limitations that we would expect agents to have in these ecologically valid tasks? The answers to these decisions would have convinced me that the task / problem space is worth exploring.

Some of the arguments and sentences could be restructured to get the main message across with less reading effort. For example, in line 26: “LiDAR-based perception system can hardly perceive the target in the occluded or long-range areas. Intuitively, with an appropriate collaboration strategy, multi-agent perception could fundamentally upgrade the perception ability over single-agent perception.” This sentence is trying to say that collectively the agents perceive more than any one given agent. And if they can pool their perception together, they would be able to make better decisions. There are quite a few sentences like this that could be simplified.

The submission uses collaboration and communication interchangeably. I suggest differentiating between them and making the writing a bit more consistent.

**New benchmark: First, why is a new task required? Why not use the V2V-Sim [31] or use AirSim-MAP or MobileNet 40 [17, 18]? Since this method directly compares against these other existing methods on similar task setup, why not evaluate on the benchmarks that have already been introduced?

The second problem is the lack of rigorous comparison to previous work. Why can’t we use the previous methods [17, 18, 31] and combine it with the spatial matrix attention contribution of this submission? I might have missed it but I didn’t see such a baseline. Similarly, why not use the knowledge distillation component with the previous methods? I know that table 3 removed both the knowledge distillation component and the maxtrix weight component from the current model. But those components can also be added to the previous methods. If it is not trivially to incorporate these contributions to existing work, perhaps add a justification for that in the text?


**Time Spent Reviewing:**

5

---

> ### Author Response · Authors · 2021-08-10
> **Thanks for your constructive comments**
>
> ### 1. Motivation of the dataset and benchmark
> We agree with the reviewer that we indeed need to illustrate the research background, and clarify the motivation for our dataset and benchmark.
>
> **1.1 Background**
>
> As one of the most exciting engineering projects of the modern world, autonomous driving is an aspiration for many researchers and engineers across generations. Although significant progress has been made, there remain many open challenges in designing a practical autonomous system that can achieve the goal of full self-driving. Among those challenges, the occlusion issue and the long-range issue in the perception module are particularly challenging and troublesome; and the resulting perception failures could cause significant cascading failures for the subsequent prediction module, planning module and control module. However, **for a single autonomous vehicle, its visibility is fixed at a moment; and the occlusion and the long-range issues are thus inevitable.**
>
> To fundamentally solve this issue in autonomous driving, we need to consider multi-agent collaborative perception, which could leverage the latest vehicle-to-vehicle (V2V) communication techniques to achieve a more comprehensive visibility and improve the perception ability. **According to the estimation of the U.S. NHTSA, there would be a minimum of 13% reduction in traffic accidents if a V2V system were implemented, which means 439,000 fewer crashes every year.**
>
> Recently, tons of research works from a communication perspective are emerging to set up a reliable V2V infrastructure; however, only a few research works from a machine learning perspective have studied the collaboration strategy towards a better perception ability based on the V2V infrastructure so far. Most works about perception are focusing on single-agent object detection and semantic segmentation; and most open perception datasets, including KITTI [8] and nuScenes [2], are all captured by single -- rather than multiple -- vehicles. Indeed, When2com, who2com, V2VNet are pioneer works on multi-agent collaborative perception. However, when2com and who2com consider 2D semantic segmentation in the setting of a collaborative group of drones, which has a clearly different objective from our goal. **While V2VNet considers the same setting with our work, neither its code nor its dataset is publicly available.** Due to this background, we are motivated to create a publicly available dataset for V2V-based collaborative perception and propose a novel collaboration strategy towards an improved ability for 3D object detection.
>
> **1.2 Motivation**
>
> * Regarding the dataset, **our motivation is to create a new and comprehensive multi-agent collaborative perception dataset, which might be the first and the only publicly available dataset for V2V-based collaborative perception.** Such a dataset would allow the research on the collaboration strategy among vehicles to achieve a more precise and reliable perception ability. This might fundamentally benefit autonomous driving, intelligent transportation systems, smart cities and many other related projects.
>
> * Regarding the benchmark, we employ the most crucial perception tasks in autonomous driving, i.e., 3D object detection, which has been extensively studied because vehicles require reliable detections to generate safe decisions. To evaluate the performance, we follow the same evaluation protocol of 3D object detection in the single-agent scenario, i.e., average precision (AP) at two thresholds (0.5 & 0.7), except that we can utilize the information shared by the neighboring vehicles while the single-agent detection cannot have access to such information. To evaluate the bandwidth, we use the straightforward communication volume. It is also possible to include the related metrics from the communication community, which is beyond the scope of the proposed research work.
>
> ***
>
> ### 2. Why not use the existing datasets?
> Our specific aim is to study V2V-based collaborative perception for autonomous driving. **The reason we cannot use the existing datasets is that there is no publicly available dataset for such a study.**
>
> * **AirSim-MAP** in When2com/who2com [17] only supports 2D semantic segmentation in a setting of a collaborative group of drones. It does not include 3D perception, such as 3D object detection, in the driving scenario.
>
> * **ModelNet 40** [17] targets shape classification based on multiple 2D image views that cannot support 3D perception in the driving scenario.
>
> * **V2V-Sim** [31]  is the most related dataset, which is developed by Uber ATG, a well-known company focusing on autonomous driving. However, V2V-Sim is not publicly available.
>
> Therefore, we have to build a dataset based on CARLA-SUMO co-simulation. Note that our dataset will be public and we believe our open-source contributions can better promote the development of multi-agent collaborative perception.
>
> ***
>
> ### 3. Comparison to two additional baselines on top of previous methods
>
> * **Incorporating the matrix-valued edge weight into the previous methods.**
> Unfortunately, the limited rebuttal period does not permit us to finish all the related experiments. Because incorporating our matrix-valued edge weight into the previous methods (Who2com, When2com and V2VNet) needs to significantly modify their original implementations and train the models from scratch, which is non-trivial. We will try to finish these experiments and reflect the results in the revised paper. Intuitively, the advantages of our method over the previous methods have been discussed in L210-L217. One key difference of our method from the previous methods is that we propose matrix-valued edge weight while the previous methods utilize a scalar attention weight for each agent: V2VNet calculates the mean of the features from neighboring agents, and When2com and Who2com utilize the attention mechanism to compute a correlation score between two agents. Please also note that, we have quantitatively validated the effectiveness of the matrix-valued edge weight in Table 3: scalar-based weighted average (56.0/50.6) -> matrix-based weighted average (60.2/54.0). In addition, Figure 5 could clearly show the effectiveness of our method qualitatively.
>
> * **Incorporating the knowledge distillation into the previous methods.**
> As the reviewer suggested, we conduct the experiments that incorporates the knowledge distillation into the previous V2VNet and When2com. We have found that the performance is largely degraded for V2VNet when equipped with distillation: (57.0/49.1 -> 47.1/42.6). The reason for this might be three-folds. First, the KD loss might influence the optimization of the convolutional gated recurrent unit (ConvGRU)-based node update module in V2VNet. Since V2VNet uses a sophisticated design and multiple iterations to update the node features, it might be hard for KD to optimize a complex architecture. Second, in the collaboration module of V2VNet, the edge weight is a single scalar across all the spatial locations, which might give very limited capacity for KD to teach. In contrast, our collaboration graph uses a matrix-valued edge weight, which provides sufficient capacity for updating. Note that when our model adopts scalar-valued edge weight, the improvements by using KD are also very limited; see Table 3. Third, the current result of V2VNet with distillation is quite preliminary. We do not have enough time to find an appropriate hyperparameter during the limited rebuttal period. Due to the similar reason, the performance for When2com based on the attention mechanism is also slightly decreased (46.7/42.4 -> 46.7/42.0). However, we believe that the proposed method are useful in general; and with appropriate modifications, both V2VNet and When2com would benefit from the distillation technique.
>
> ***
>
> ### 4. Applicable tasks
>
> Our method is effective in various 3D perception tasks, such as 3D object detection, 3D object tracking, 3D point cloud segmentation, and the subsequent motion prediction. The intuition is that each LiDAR has limited sensing ability; while multiple LiDARs together could provide more evidence of the scene. An effective collaboration strategy would be able to generally empower multiple agents to see better, further and through occlusion, which does not limit to a specific application.
>
> ***
>
> ### 5. Discussion around the dataset
>
> There is a detailed introduction of our dataset in Appendix A. Although we only use one town, there are more than twenty intersections which are used for dataset generation. For realistic traffic simulation, we use SUMO to produce random routes for five hundred vehicles and make them roam around. Several generated scenes are visualized in Figure 10, and we randomly select agents for collaboration to ensure the diversity of V2V scenarios, e.g., platoon/head-on/abreast, low/high-density traffic, low/high-overlap observation regions between agents, etc. Note that this dataset will be enriched progressively, and more scenarios like adverse weather and high-speed driving will be included and evaluated in our follow-up works. Regarding the benchmark, we have focused on overall detection performance and communication bandwidth. More detailed evaluation metrics could be proposed as our reviewer suggested. We really thank you for your constructive feedback.

---

### Decision · Program_Chairs · 2021-09-27

**Decision:**

Accept (Poster)

**Comment:**

This paper proposes a new architecture for collaborative multi-agent perception problem. The core idea is to train a (privileged) teacher which receives a holistic-view point cloud from all agents and distill the teacher's feature map to individual agents to encourage each agent to incorporate the other agents' information without direct access to their information. The paper also introduces a large-scale multi-agent 3D object detection dataset and show that the proposed method outperforms the baselines with much less communication.

All of the reviewers agreed that the proposed architecture is reasonable and novel, and the overall results look good as well. In addition, the new dataset introduced by the paper would be valuable to the multi-agent perception community. In the meantime, there was a concern that the main compression benefit comes from 1x1 conv rather than the main idea of the paper. But, the authors promised to revise the paper so that the contribution of each idea becomes clear, and the reviewers acknowledged that 1x1 conv is also a part of the proposed architecture. Another concern was that it was not crystal clear why the proposed method is not evaluated on 2D datasets, where the baselines were evaluated. The authors clarified that the proposed architecture is specifically designed for 3D perception, though some of the ideas could be generally applicable. I encourage the authors to clarify this in the camera-ready version.

Assuming that the authors will reflect these comments, I'd recommend to accept the paper.